# GNNXEMPLAR: Exemplars to Explanations - Natural Language Rules for Global GNN Interpretability

**Burouj Armgaan**
Dept. of CSE, IIT Delhi
csz228001@iitd.ac.in

**Eshan Jain**
Dept. of CSE, IIT Delhi
eshan292@gmail.com

**Harsh Pandey**
Fujitsu Research of India, Bangalore
Harsh.p@fujitsu.com

**Mahesh Chandran**
Fujitsu Research of India, Bangalore
Mahesh.Chandran@fujitsu.com

**Sayan Ranu**
Dept. of CSE and Yardi ScAI, IIT Delhi
sayanranu@iitd.ac.in

## Abstract

Graph Neural Networks (GNNs) are widely used for node classification, yet their opaque decision-making limits trust and adoption. While local explanations offer insights into individual predictions, global explanation methods, those that characterize an entire class, remain underdeveloped. Existing global explainers rely on motif discovery in small graphs, an approach that breaks down in large, real-world settings where subgraph repetition is rare, node attributes are high-dimensional, and predictions arise from complex structure-attribute interactions. We propose GNNXEMPLAR, a novel global explainer inspired from *Exemplar Theory* from cognitive science. GNNXEMPLAR identifies representative nodes in the GNN embedding space, exemplars, and explains predictions using natural language rules derived from their neighborhoods. Exemplar selection is framed as a coverage maximization problem over reverse $k$-nearest neighbors, for which we provide an efficient greedy approximation. To derive interpretable rules, we employ a self-refining prompt strategy using large language models (LLMs). Experiments across diverse benchmarks show that GNNXEMPLAR significantly outperforms existing methods in fidelity, scalability, and human interpretability, as validated by a user study with 60 participants.

## 1 Introduction and Related Works

Node classification using Graph Neural Networks (GNNs) has found wide-ranging applications across diverse domains, including protein function prediction [13, 30], user profiling [47, 11], and fraud detection [22]. Despite their success, GNNs, like other deep learning models, are often regarded as black boxes: they achieve high performance, but the reasoning behind their predictions remains largely opaque. To address this, GNN *explainers* aim to illuminate the decision-making process by identifying the substructures and features in the input graph that the model relies on for its predictions [1, 2, 48, 24, 39, 18, 42, 16, 17].

### 1.1 Existing Works and Open Challenges

Existing methods for GNN explanation can be broadly categorized into two types: local and global. Fig. 4 in the Appendix provides the detailed taxonomy.

1. **Local explanations** focus on individual predictions. They aim to explain why a specific input graph (or node) received a particular label. While informative, these explanations are often instance-specific and do not generalize beyond the particular case being analyzed.
2. **Global explanations**, in contrast, seek to uncover general patterns that apply across multiple instances. They aim to explain the conditions under which a GNN assigns any input to a particular class, producing a single, interpretable explanation per class.

While local explanation techniques are relatively mature, global GNN explainers remain an emerging area of research, as evident from Fig. 4. In this work, we advance this direction by proposing a global explanation framework for node classification based on *exemplar theory* [28].

Exemplar theory, rooted in cognitive psychology, explains how humans categorize objects and ideas. It posits that individuals make category judgments by comparing new stimuli with previously encountered instances, called *exemplars*, stored in memory. These exemplars are not arbitrary; they tend to represent *typical* members of a category [28]. A new stimulus is assigned to a category based on the number and degree of similarities it shares with exemplars from that category.

We adapt this idea to GNN-based node classification by treating certain representative nodes in the embedding space as *exemplars*: nodes that encapsulate the characteristics of many others in the same class. For each exemplar, we extract an interpretable signature describing the distribution of features and/or structural patterns that characterizes the exemplar and the population of nodes it represents in the embedding space. Then, for any unseen node, we explain the GNN's prediction by referencing the signature of the exemplar it most closely resembles. This approach enables global yet instance-relevant explanations grounded in learned graph structure and semantics.

**Open Challenges:** Existing global GNN explainers [50, 42, 51, 45, 1, 2, 25] have primarily targeted graph classification tasks on small-scale datasets such as molecular graphs. These methods typically aim to identify recurring substructures, commonly referred to as motifs or concepts, that the GNN being explained detects to classify graphs [50, 42, 51, 45, 25] Some explainers [1, 2] go further by constructing the Boolean logic rules over motifs that aligns with the model's predictions. However, these motif-based explanation strategies face significant challenges when extended to node classification in large, real-world graphs with rich node attributes (Ex. citation and linked documents, financial networks, etc.)

- **Attribute-Topology Interaction:** In large graphs, GNN predictions are typically a complex function of both the graph structure and high-dimensional node attributes. This makes motif discovery difficult because the classical definition of a motif, grounded in graph isomorphism, handles only discrete node labels. Moreover, in real-world graphs, the repetition of the exact same subgraph with identical node attributes is rare, making the very notion of a motif ambiguous. This calls for a shift from exact, symbolic subgraph matching to an approximate space where repetitions of structurally and semantically similar patterns can be meaningfully observed. We address this by identifying recurring combinations of structure and attributes not in the raw input graph but in the GNN embedding space by locating dense neighborhoods.
- **Computational Complexity:** Motif discovery involves solving the *subgraph isomorphism* problem, which is NP-hard, making them prohibitively expensive on large graphs. The problem is further exacerbated when accounting for both topology and node attributes. Consequently, as we will show later in § 4, existing global GNN explainers often fail to scale on large graphs.
- **Cognitive and Visual Overload:** In small graphs, like molecules, motifs (e.g., functional groups) are compact and human-interpretable. However, in large real-world graphs, even the 2-hop neighborhood of a node can include hundreds or thousands of nodes. Visualizing or interpreting such patterns quickly becomes unwieldy and exceeds human cognitive limits.

## 1.2 Contributions

In this work, we present GNNXEMPLAR, which addresses the limitations outlined above.

- **Problem formulation:** Effective explanations of GNN predictions must satisfy two key criteria: (1) they should be *faithful*, meaning they accurately reflect the model's decision-making process, and (2) they should be *interpretable*, allowing humans to understand the reasoning behind predictions despite the underlying complexity of the graph modality. To address these dual objectives, we take inspiration from *Exemplar Theory* [28]. Exemplar theory posits that humans categorize new instances by comparing them to representative examples (exemplars) previously encountered. By adapting this principle, we explain the GNN's prediction for a node by referencing similar, representative nodes in the embedding space — its exemplars. Furthermore, to enhance interpretability, we move beyond subgraph visualizations which are inherently ineffective in large, dense graphs due to their complexity and scale. Instead, we distill the defining characteristics of each exemplar and its associated population into *textual explanations*. This shift to natural language enables more accessible and cognitively manageable insights into the GNN's decision-making process. We validate this claim through a user survey (§ 4.7).

- **Novel methodology:** We develop GNNXEMPLAR, which integrates several innovative components. First, exemplar identification is cast as a coverage maximization problem over reverse $k$-nearest neighbor relationships. We prove that this problem is NP-hard and submodular, and accordingly propose a greedy algorithm with a $(1 - \frac{1}{e})$ approximation guarantee to the optimal solution. To uncover the signature characteristics of each exemplar and the population it represents, we leverage LLMs to iteratively propose and refine interpretable logical rules that are consistent with the GNN's predictions.
- **Empirical analysis:** We conduct extensive experiments on a diverse suite of homophilous and heterophilous graphs. Our analysis reveals that: (1) existing GNN explainers are inadequate for node classification on large graphs with complex node attributes; (2) GNNXEMPLAR provides high-fidelity explanations of GNN predictions; and (3) the text-based explanations are preferred over subgraph visualization, as validated through a user study involving 60 participants.

## 2 Preliminaries and Problem Formulation

**Definition 1** (Graph). *A graph is defined as $\mathcal{G} = (\mathcal{V}, \mathcal{E}, \boldsymbol{X})$ over a node set $\mathcal{V}$, edge set $\mathcal{E} = \{(u, v) \mid u, v \in \mathcal{V}\}$ and node attributes $\boldsymbol{X} = \{\mathbf{x}_v \mid v \in \mathcal{V}\}$ where $\mathbf{x}_v \in \mathbb{R}^d$ is the set of features characterizing each node.*

**Definition 2** (Node Classification). *In node classification, we are given a single input graph $\mathcal{G} = (\mathcal{V}, \mathcal{E}, \mathbf{X})$, where a subset of nodes $\mathcal{V}_{tr} \subset \mathcal{V}$ is associated with known class labels from the set $\{y_1, \cdots, y_c\}$. The objective is to train a GNN model $\Phi$ such that, for any node $v \in \mathcal{V} \setminus \mathcal{V}_{tr}$, the prediction error of its class label is minimized.*

Error may be measured using known metrics such as cross-entropy loss, negative log-likelihood, etc.

**Definition 3** (Exemplars). *Exemplars refer to representative nodes within the graph that embody prototypical structural and attribute characteristics shared by many other nodes in the same class. In the context of node classification, these exemplars serve as anchors for human-understandable explanations. Each exemplar, $e$, is associated with a subset of training nodes, $R_e \subseteq \mathcal{V}_{tr}$, which represents the population that $e$ exemplifies. A node $e \in \mathcal{V}_{tr}$ is called an exemplar for the class $c \in \mathcal{Y}$ if $|R_e| \geq \tau$, where*

$$R_e = \left\{ v \in \mathcal{V}_{tr} \;\middle|\; \begin{array}{l} \Phi(e) = c, \\ \Phi(v) = c, \\ d(z_e, z_v) \leq \delta \end{array} \right\}$$

*where $z_v \in \mathbb{R}^d$ is the GNN-learned embedding of node $v$, $d(z_e, z_v)$ is some distance function, $\delta \in \mathbb{R}$ is a distance threshold, and $\tau \in \mathbb{N}$ is a minimum population size hyperparameter.*

As we will see later, the thresholds will automatically be discovered from the data by exploiting the $k$-nearest neighbor relationship along with a greedy selection algorithm.

**Definition 4** (Exemplar Signature). *The signature of exemplar $e$, denoted as $\sigma_e$, is a boolean formula composed of interpretable conditions over the structural and attribute properties of the local neighborhood of $e$. This formula is constructed such that its truth value approximates $\Phi$'s predictions over $R_e$, i.e., with high probability $\forall v \in R_e, : \sigma_e(v) = \Phi(v)$. The signature $\sigma_e$ of exemplar $e$ from class $c$ is a boolean function $f(\pi_1(v), \pi_2(v), \ldots, \pi_k(v)) \in \{0, 1\}$, such that, with a high probability, $\forall v \in R_e, \sigma_e(v) = 1$, iff $\Phi(v) = c$. Here, $\pi_i$ are the predicates that evaluate to Boolean conditions over interpretable properties such as:*

- *Node features (e.g., $x_v[\texttt{age}] > 30$)*
- *Local structure (e.g., $\texttt{degree}(v) \geq 3$)*
- *Neighborhood composition (e.g., $\texttt{fractionClass}(v, \texttt{hop} = 1, c') > 0.6$)*

For instance, an exemplar of fraudulent nodes in a transaction graph, might be an account that makes frequent, low-value transfers to many recently created accounts that show no further activity. Assuming each node has attributes indicating the average transaction value and average frequency of transactions per week, its signature could be a rule like: *"Has an average transaction amount below \$100, performs more than 10 transactions per week, and is dissimilar in transaction frequency from the majority of its neighbors"*

**Problem 1** (Global GNN Explanation through Exemplar Signatures). *Let $\Phi$ be a trained GNN that assigns to each node $v \in \mathcal{V}$ a class label from $\{y_1, \cdots, y_c\}$. For each class $y_i$, let $\mathcal{E}_i \subseteq \mathcal{V}$ denote a*

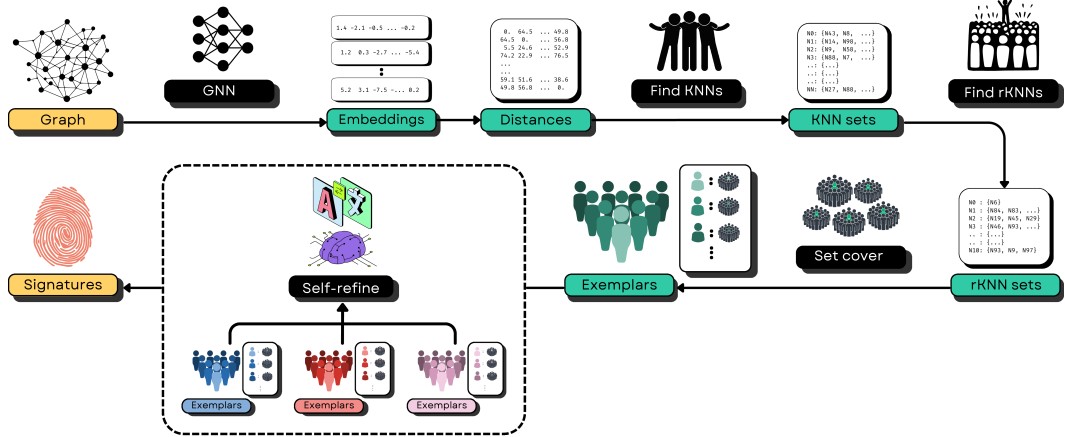

Figure 1: The pipeline of GNNXEMPLAR.

*selected set of exemplar nodes that are representative of class $y_i$, and let $\sigma_e$ be the boolean signature associated with exemplar $e \in \mathcal{E}_i$.*

*The goal is to construct a global explanation in the form of a Boolean formula $f_i$ for each class $y_i$, such that:*

$$f_i(v) = \bigvee_{e \in \mathcal{E}_i} \sigma_e(v)$$

*where, $\forall v \in \mathcal{V}_{tr}$, $\Phi(v) = y_i \Leftrightarrow f_i(v) = \text{TRUE}$. Each $f_i$ serves as the global explanation for class $y_i$, built by aggregating exemplar-level signatures via logical OR. Each exemplar signature is free to use all Boolean operators.*

The formulation of explanation through exemplars presents two central challenges:

1. *How do we identify an optimal set of exemplars?* Exemplar selection is a combinatorial optimization problem over the space of all nodes in the graph. Choosing too many exemplars compromises both interpretability and computational efficiency. A small, well-chosen exemplar set enables concise global explanations, as the final explanation takes the form of a disjunction over exemplar-specific rules.

2. *How do we derive boolean logic signatures for each exemplar?* The space of candidate logical expressions, composed of node attributes and structural features, is exponentially large, rendering exhaustive search intractable. Additionally, these rules must strike a balance between fidelity to the GNN's predictions and human interpretability. A further challenge arises from the need to express these logical rules as natural language descriptions requiring semantic precision.

## 3 Methodology

Fig. 1 presents the pipeline of GNNXEMPLAR. There are two distinct components: first, we identify a budget-constrained set of exemplars, and next, we extract boolean rules, expressed in natural language, through iterative self-refining using LLM. We next detail each of these steps.

### 3.1 Exemplar Identification.

GNNXEMPLAR aims to identify a set of $b$ *exemplar* nodes that optimally represent the full training set; $b$ is a tunable hyper-parameter balancing complexity of the explanation with higher expressivity. This selection process is guided by two critical criteria:

- **Representativeness:** Each exemplar node should be close to a large number of other nodes sharing the same GNN-*predicted* class label in the embedding space, thereby capturing common structural and attribute patterns learned by the model. We quantify representativeness using the notion of *reverse $k$-nearest neighbors* (§ 3.2) in the GNN embedding space. Specifically, if node $v$ appears frequently in the $k$-NN sets of other nodes with the same *predicted* label, it is considered representative of those nodes' learned semantics. Such nodes are prioritized for inclusion in the exemplar set, as they enable coverage of large regions of the embedding space and are likely to approximate the model's behavior over similar instances.

- **Diversity:** The selected exemplars should be well spread out in the GNN embedding space to ensure a wide range of model behaviors are covered (§ 3.3).

By jointly optimizing for representativeness and diversity, GNNXEMPLAR ensures that the selected exemplars collectively span a broad range of the model's learned representations.

## 3.2 Quantifying Representativeness through Reverse $k$-NN

The representative power of a node is defined as follows.

**Definition 5** (Reverse $k$-NN and Representative Power). *Let $\mathbf{h}_v$ denote the GNN embedding of node $v$. The $k$ nearest neighbors of node $v$, denoted as $k$-NN$(v)$, are the $k$ nodes from the train set with embeddings closest to $\mathbf{h}_v$ and sharing the same GNN predicted class label (we use $L_2$ distance, but other distances may also be used). The reverse $k$-NN of node $v$ is the set of nodes for which $v$ appears in their $k$-NN set. Formally,*

$$Rev\text{-}k\text{-}NN(v) = \{u \in \mathcal{V}_{tr} \mid v \in k\text{-}NN(u), \Phi(v) = \Phi(u)\}$$

*The representative power of node $v$ is then defined as*

$$\Pi(v) = \frac{|Rev\text{-}k\text{-}NN(v)|}{|\{u \in \mathcal{V}_{tr} \mid \Phi(v) = \Phi(u)\}|}$$

A high value of $\Pi(v)$ indicates that node $v$ frequently appears in the $k$-NN sets of many other nodes, implying it resides in a dense region of the embedding space and captures shared representational characteristics. Thus, it is a strong candidate for inclusion in the exemplar set.

### 3.2.1 Sampling for Scalable Computation of Reverse $k$-NN

Computing the $k$ nearest neighbors for each node consumes $\mathcal{O}(n \log k) \approx \mathcal{O}(n)$ time, since $k \ll n$ and $n = |\mathcal{V}_{tr}|$ is the number of nodes. Thus, computing $k$-NN for all nodes and then constructing the Rev-$k$-NN requires $\mathcal{O}(n^2)$ time, which becomes prohibitively expensive on large-scale graphs.

To address this, we adopt a sampling-based strategy that enables scalable approximation of reverse $k$-NN with theoretical [12]. Let $\mathbb{S} \subset \mathcal{V}_{tr}$ be a uniformly sampled subset of $z \ll n$ nodes. We compute the $k$-NN only for nodes in $\mathbb{S}$, which reduces the computational cost to $\mathcal{O}(zn)$.

We then define the approximate reverse $k$-NN of a node $v \in \mathcal{V}_{tr}$ as $\widetilde{Rev\text{-}k\text{-}NN}(v) = \{u \in \mathbb{S} \mid v \in k\text{-}NN(u), \Phi(v) = \Phi(u)\}$. Hence, the approximate representative power of $v$ is

$$\widetilde{\Pi}(v) = \frac{|\widetilde{Rev\text{-}k\text{-}NN}(v)|}{|\{u \in \mathbb{S} \mid \Phi(v) = \Phi(u)\}|}$$

Here, $\widetilde{\Pi}(v)$ estimates how broadly node $v$ is retrieved as a nearest neighbor across the sample set. The sample size $z$ controls the trade-off between accuracy and efficiency. Leveraging Chernoff bounds, we establish that even a small $z$ yields high-confidence approximations:

**Lemma 1.** *Given an error threshold $\theta$ and confidence level $1 - \delta$, it suffices to sample $z \geq \frac{\ln\left(\frac{2}{\delta}\right)(2+\theta)}{\theta^2}$ nodes to ensure that for any $v \in \mathcal{V}_{tr}$, $P\left(\left|\widetilde{\Pi}(v) - \Pi(v)\right| \leq \theta\right) \geq 1 - \delta$.*

PROOF. See App. A.3.

This result has two key implications:

- The required number of samples is independent of the number of nodes in the train set, i.e., $|\mathcal{V}_{tr}|$.
- Since $z$ scales logarithmically with $\ln(1/\delta)$, even a small sample ensures high-probability bounds. Thus, the overall computation cost for reverse $k$-NN becomes $\mathcal{O}(n)$ instead of $\mathcal{O}(n^2)$.

While we use the notations Rev-$k$-NN $(v)$ and $\Pi(v)$ in the subsequent discussion, approximate variants via sampling may be used in practice. We evaluate the quality-efficiency trade-off in App. §A.7.

## 3.3 Coverage Maximization

We aim to identify a subset of $b$ *exemplar nodes* that collectively offer the broadest coverage over the training set in the GNN embedding space.

**Algorithm 1** Greedy Node Selection

**Require:** Graph $\mathcal{G}$, budget $b$, Rev-$k$-NN sets of nodes
**Ensure:** exemplar set $\mathbb{A}$ of size $b$
1: $\mathbb{A} \leftarrow \emptyset$
2: **while** $|\mathbb{A}| < b$ **do**
3: $\quad v^* \leftarrow \arg\max_{v \in \mathcal{V}_{tr} \setminus \mathbb{A}} \Pi(\mathbb{A} \cup \{v\}) - \Pi(\mathbb{A})$
4: $\quad \mathbb{A} \leftarrow \mathbb{A} \cup \{v^*\}$
5: **Return** $\mathbb{A}$

**Definition 6** (Exemplar Set). *Let the representative power of a set of exemplar nodes $\mathbb{A} \subseteq \mathcal{V}$ be defined as:*

$$\Pi(\mathbb{A}) = \left| \bigcup_{v \in \mathbb{A}} \textit{Rev-k-NN}(v) \right| \Big/ |\mathcal{V}_{tr}| \tag{1}$$

*Here, Rev-$k$-NN($v$) denotes the reverse $k$ nearest neighbors of node $v$ in the GNN embedding space.*

*Given a node set $\mathcal{V}$ and budget $b$, we seek a subset of train nodes $\mathbb{A}^* \subseteq \mathcal{V}_{tr}$ of size $b$ that maximizes:*

$$\mathbb{A}^* = \arg \max_{\mathbb{A} \subseteq \mathcal{V}_{tr}, |\mathbb{A}| = b} \Pi(\mathbb{A}) \tag{2}$$

**Theorem 1.** *Maximizing the representative power in Eq. 2 is NP-hard.*

PROOF. See App. A.4 for a reduction from the classic *Set Cover Problem* [5]. □

Fortunately, the objective $\Pi(\mathbb{A})$ is monotone and submodular, making it amenable to greedy approximation.

**Theorem 2.** *The greedy algorithm (Alg. 1) guarantees $\Pi(\mathbb{A}_{greedy}) \geq \left(1 - \frac{1}{e}\right) \Pi(\mathbb{A}^*)$.*

PROOF. App. A.5 shows the monotonicity and submodularity of $\Pi(\mathbb{A})$. An analogous proof for the case using the approximate Rev-$k$NN is provided in App. A.6. □

Alg. 1 iteratively selects nodes that maximize marginal gain in coverage. By exploiting the transitivity of proximity in the embedding space, this greedy strategy avoids redundancy: if two nodes share most of their reverse $k$-NN sets, selecting one reduces the marginal utility of the other. Thus, the algorithm promotes both coverage and diversity within the budget.

### 3.4 Discovery of Exemplar Signatures

Our goal is to analyze the embeddings of an exemplar node and its Rev-$k$-NN set, and derive a symbolic Boolean rule, expressed in natural language, that closely matches the GNN's predictions on this population (see Prob.1). To this end, we leverage the reasoning capabilities of large language models (LLMs), motivated by two key factors. *(1)* LLMs have demonstrated strong mathematical and causal reasoning abilities, including over graph-structured data[55, 6, 26, 49]. *(2)* Given our aim to express logical rules in natural language, LLMs are particularly well suited due to their well-established strengths in linguistic articulation. **Self-refinement paradigm:** Fig. 2 presents the pipeline of the signature discovery process of an exemplar, which leverages the *self-refine* platform [26]. We select an initial sample of *positive* and *negative* nodes. The positive sample contains nodes from the Rev-$k$-NN set, and the negative sample contains those not in the Rev-$k$-NN. Both samples are drawn uniformly at random. The sample size is a hyperparameter. These node sets are further divided into training and validation sets. Instead of asking the LLM to directly output the boolean rule in natural language, we decouple it into two steps. The LLM is first asked to generate a *python code*, that takes as input a node and its associated information, passes it through a boolean logic, and returns a `True`/`False` answer. The ideal rule returns `True` for all positive nodes and `False` for negative nodes. The rule encoded in the Python code is then translated to natural language by the LLM. Initially, the Python code is a trivial solution, such as returning `True` for any input node, which is iteratively improved through *feedback*. The iterations stop when either the accuracy of the code (rule) exceeds a certain accuracy threshold on the validation set or the number of iterations reaches an upper limit.

**Prompt specification:** The prompt includes the following components; a visual overview is provided in Fig. 10 (query), Fig. 11 (*feedback*), and Fig. 12 (*refinement*) in the Appendix.

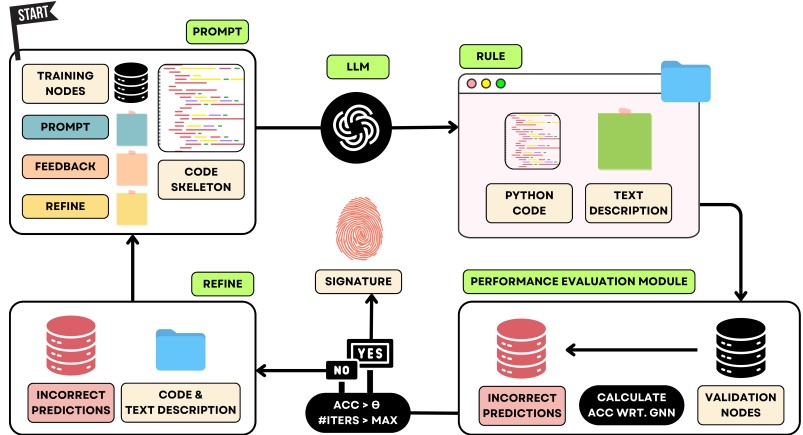

Figure 2: Pipeline of signature discovery through iterative self-refinement of LLM output.

- **Training and validation nodes:** As discussed above, we provide a set of positive and negative nodes. If the GNN being explained is $\ell$ layers deep, then the embedding of a node is a function of its $\ell$-hop neighborhood only [44]. Hence, we provide a *summary* of the $\ell$-hop neighborhoods of each node in the *positive* and *negative* samples. The summary of a node includes: (1) the attributes of the node, *(2)* the normalized frequency distribution of GNN predicted class labels for all nodes in each hop from $1$ to $\ell$, *(3)* the average $L_1$ distance per attribute between the exemplar and all nodes at each hop level from $1$ to $\ell$. While GNN class distribution signals the degree of homophily-bias learned by the GNN, the average attribute-wise distance offers insight into feature relevance in the embedding space. Specifically, if the average distances to positive and negative nodes are similar for a given attribute, it suggests that the feature contributes little to the embedding representation. Note that, since we only provide distribution-level information, the context size remains independent of the graph's density. This property is crucial for ensuring that large graphs do not exceed the LLM's context window capacity.
- **Code skeleton:** The skeleton of the python code that the LLM is supposed to fill in. The skeleton contains the function definition that takes as input the summary of a node and the output specification, which should return a boolean value.
- **Feedback:** This includes the Python code and the associated natural language rule from the last iteration and the summary of all nodes where the latest rule failed to match the GNN prediction.

## 4 Experiments

In this section, we benchmark GNNXEMPLAR and establish:
- **Limitations of Existing Explainers:** Current explainers, primarily designed for graph classification on small graphs, fail to generalize effectively to node classification in large-scale graphs.
- **High-Fidelity explanations:** GNNXEMPLAR bridges this gap by consistently achieving high fidelity across both homophilic and heterophilic graph benchmarks.
- **User Preference for Textual Explanations:** Our user survey involving 60 participants shows a statistically significant preference for textual explanations over graph-based visualizations.

### 4.1 Experimental Setup

The details of our hardware and software platform, hyper-parameters, LLM engine, training details of the black-box GNN and their accuracies are discussed in App. A.1 and App. A.2. Our codebase is shared at https://github.com/idea-iitd/GnnXemplar.git.

**Datasets:** Table 1 presents the $8$ benchmark datasets we use. Wherever available, we adopt the standard train/validation/test splits from PyTorch Geometric or the original data releases, preserving class balance. We train a GAT for TAGCora and GCN for the rest.

**Baselines:** As discussed in § 1, there are no existing explainers designed to handle node classification on large graphs. Hence, we adapt state-of-the-art explainers originally developed for graph classification. To enable a fair comparison, we reframe the node classification task as a graph classification problem by extracting the $\ell$-hop subgraph around each target node and assigning the node's label to the entire subgraph; $\ell$ denotes the number of layers in the GNN. We compare against the following baselines:

Table 1: Dataset statistics. Detailed descriptions of each dataset are in App.A.9

| Homophilous | | | | | Heterophilous | | | | |
|---|---|---|---|---|---|---|---|---|---|
| **Name** | **#nodes** | **#edges** | **#features** | **#classes** | **Name** | **#nodes** | **#edges** | **#features** | **#classes** |
| TAGCora [21] | $2,708$ | $10,556$ | 1433 | 7 | Amazon-ratings [32] | $24,492$ | $93,050$ | 300 | 5 |
| Citeseer [10] | $3,327$ | $9,104$ | 3703 | 6 | Minesweeper [32] | $10,000$ | $39,402$ | 7 | 2 |
| WikiCS [29] | $11,701$ | $216,123$ | 300 | 10 | Questions [32] | $48,921$ | $153,540$ | 301 | 2 |
| ogbn-arxiv [14] | $169,343$ | $1,166,243$ | 128 | 40 | BA-Shapes [48] | 300 | 4110 | 0 | 4 |

- **GNNInterpreter** [42]: A generative model trained to synthesize class-representative graphs using reinforcement learning.
- **GCNeuron** [45]: A neuron-level explainer that identifies high-level subgraph concepts that activate neurons within a GNN.
- **GLGExplainer** [2]: The only prior global logical explainer that fits a Boolean formula to GNN outputs by clustering local explanation subgraphs (e.g., PGExplainer [24]) around prototypes and learning a formula over those prototypes using ELEN [4].

Since both GNNInterpreter and GCNeuron identify representative subgraphs without generating explicit Boolean rules, we construct a rule by taking a logical OR over all identified subgraphs for each class label. We do not compare with GraphTrail [1] since it only considers graphs with discrete node labels. MAGE [50] is also omitted since it is specific to molecular graphs.

**Metrics:** We evaluate each explainer by applying its generated Boolean formula to the test set of each dataset. The alignment between the formula and the GNN's predictions is quantified using the following metrics: *Fidelity* (the proportion of test nodes where the formula's output matches the GNN's prediction), *Precision*, *Recall*, and *F1-score*.

## 4.2 Fidelity

Table 2 reports the fidelity of each explainer on node classification tasks. Precision, recall and F-score for the same experiments are reported in Tables 6, 7 and 8 in the Appendix. Several key observations emerge. **(1)** GNNXEMPLAR consistently achieves the highest fidelity, establishing a new benchmark for global explanation in node classification. **(2)** Explainers for graph classification demonstrate brittle performance when adapted to node classification, confirming that exact subgraph repetition is rare in real-world graphs. **(3)** motif-based explainers reliant on subgraph isomorphism fails to scale on large graphs. We elaborate below.

**NAs:** The NA cases occur when an explanation is generated but cannot be applied or evaluated on any specific node (or neighborhood subgraph) instance. This limitation arises from the reliance of GNNInterpreter on subgraph-based rules. Its only mode of application is through subgraph isomorphism, which severely constrains its generalizability. Subgraph isomorphism can only function when node features are either absent or purely discrete (e.g., scalars). With continuous or high-dimensional node features, determining whether two nodes, or their neighborhoods, are "identical" becomes ill-defined, rendering the explanation unusable in most real-world settings.

**NFs:** NF stands for "No Formula Generated," which is encountered in GLGExplainer. GLGExplainer first trains a surrogate model to mimic the black-box GNN and then attempts to distill this surrogate into interpretable boolean logic. However, we find that in several datasets, either the surrogate model fails to approximate the GNN, or the boolean distillation step breaks down. In both scenarios, GLGExplainer outputs an empty string. This failure indicates that GLGExplainer is brittle when faced with noisy, irregular, or highly entangled graph patterns.

**Inferior Baseline Performance.** Even when the baselines produce actionable rules, all three yield significantly lower fidelity. This is primarily because they rely on identifying common subgraph patterns per class, an assumption that rarely holds in complex, real-world graphs. In contrast, GNNXEMPLAR is grounded in exemplar theory and computes distributional distances to strategically selected exemplars in the GNN embedding space. This approach avoids subgraph isomorphism while remaining faithfully aligned with the GNN's predictive behavior. **Scalability:** Both GLGEXPLAINER and GCNEURON frequently encounter out-of-memory (OOM) errors on large or dense graphs, with GLGEXPLAINER failing because of its reliance on subgraph-isomorphism and GCNEURON due to exhaustive neuron-activation pattern enumeration. In contrast, GNNXEMPLAR is explicitly designed to avoid subgraph isomorphism and scales gracefully.

Table 2: Fidelity of the GNNXEMPLAR and the various baselines. OOM indicates out-of-memory. NA indicates "Not Applicable", and NF indicates the case where the algorithm failed to generate a formula. GNNInterpreter assumes discrete node attributes only, and hence, for datasets with continuous attributed notes, it is not applicable.

| | Homophilous | | | | Heterophilous | | | |
| --- | --- | --- | --- | --- | --- | --- | --- | --- |
| | TAGCora | Citeseer | WikiCS | arxiv | Amazon-R | Questions | Minesweeper | BA-Shapes |
| GNNInterpreter | NA | $0.50 \pm 0.0$ | NA | NA | NA | NA | $0.50 \pm 0.0$ | $0.47 \pm 0.0$ |
| GCNeuron | $0.51 \pm 0.0$ | $0.50 \pm 0.0$ | OOM | OOM | $0.56 \pm 0.0$ | OOM | $0.54 \pm 0.0$ | $0.50 \pm 0.0$ |
| GLGExplainer | NF | NF | OOM | OOM | NF | OOM | $0.22 \pm 0.07$ | $0.30 \pm 0.09$ |
| GNNXEMPLAR | $0.83 \pm 0.01$ | $0.92 \pm 0.03$ | $0.78 \pm 0.01$ | $0.84 \pm 0.01$ | $0.82 \pm 0.01$ | $0.92 \pm 0.01$ | $0.86 \pm 0.02$ | $0.93 \pm 0.00$ |

## 4.3 Ablation Study

To quantify the contributions of our two key innovations, exemplar selection through Rev-$k$-NN and iterative self-refinement using LLM, we conduct two controlled ablations.

**Rev-$k$-NN vs. Random Exemplar Selection:** Here, we replace our Rev-$k$-NN based exemplar selection with an equal-sized set of randomly sampled nodes (exemplars) per class. This change yields a noticeable fidelity decrease, as random prototypes fail to capture the dense, semantically coherent neighborhoods that Rev-$k$-NN provides (see Fig. 3-Left subplot).

**Self-Refinement vs. Zero-Shot:** In the zero-shot variant, we bypass the structure self-refining pipeline discussed in § 3.4 and directly ask the LLM to predict the rule (signature) in one go. Ablation results in Fig. 3 (Right subplot) demonstrate that this zero-shot approach yields noticeably lower fidelity and higher variance on every dataset. This reveals that the full self-refinement pipeline systematically identifies misclassified instances and adjusts the rules accordingly, producing significantly more accurate and robust explanations with reduced variance across runs.

## 4.4 Impact of Parameters and GNN architectures

App. A.7 studies the impact of parameters $k$ in Rev-$k$-NN, sample size in Rev-$k$-NN approximation, and the training-set sample size in self-refine. App. A.8 evaluates robustness to GNN architectures.

## 4.5 Case Study: Visual Analysis of Rules

To demonstrate the interpretability and domain-alignment of our global explanations, we show representative rule sets from TAGCora (homophilous) (Fig. 14) and BAShapes (heterphilous) (Fig. 15).

**TAGCora (Fig. 14)** In the TAGCora citation network, topic (class) assignments are driven by both textual features and neighborhood context. For instance, *Neural Networks* papers are identified by the presence of domain-specific terms like "backpropagation" and "activation functions". Furthermore, the rule also discovers homophillic associations with other "Neural Network" papers among 1 and 2-hop neighbors. The rule further comments that the presence of ICA and HMM words in the abstracts of the neighbors further reinforces the class assignment.

**BAShapes (Fig. 15)** On the synthetic BAShapes graph, membership in each geometric role is determined solely by local connectivity patterns. GNNXEMPLAR is successfully able to identify these purely structural and heterophilic associations, such as nodes belonging to the class "not in a house" have at least six "non-house" neighbors and minimal ties to other roles.

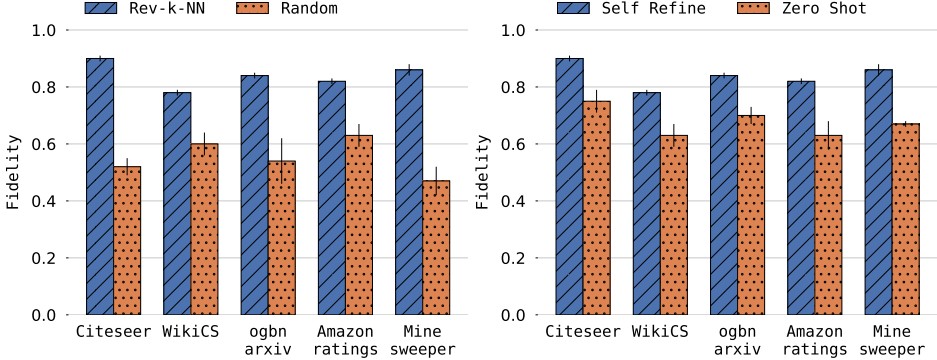

Figure 3: Ablation study on GNNXEMPLAR. **Left:** Impact of Rev-$k$-NN based exemplar selection. **Right:** Impact of self-refinement strategy vs. zero-shot rule discovery.

Table 3: Results of one-sided binomial tests assessing user preference for text-based explanations over subgraph-based explanations across five A/B test questions. Bolded $p$-values indicate a statistically significant preference for the text modality ($p < 0.05$).

| | Q1 | Q2 | Q3 | Q4 | Q5 | Total |
|---|---|---|---|---|---|---|
| **Dataset** | TAGCora | Questions | Questions | BAShapes | BAShapes | – |
| **Prefer text** | 43 | 41 | 36 | 42 | 38 | 200 |
| **Prefer subgraph** | 17 | 19 | 24 | 18 | 22 | 100 |
| $p$-**value** | **0.0005** | **0.0031** | 0.0775 | **0.0013** | **0.0259** | **< 0.0001** |

### 4.6 Application: Diagnostic Power in Failure Cases

In Questions, active users exhibit homophilous behavior–they tend to connect to other active users. Inactive users, in contrast, behave heterophilously by connecting primarily to active users. Interestingly, our explanation rule for the inactive class does not reflect this expected heterophily. Instead, it points toward a homophilous pattern even for inactive users. At first glance, this seems contradictory: how can an explanation be so seemingly incorrect and still achieve high fidelity (see Table 2)?

To investigate, we visualized 2-hop neighborhoods of inactive nodes: once with ground-truth labels, and once with GNN predictions (see example in Fig. 5). The results were revealing: the GNN predicts even the neighbors of inactive nodes as inactive, indicating that it has learned an incorrect homophilous pattern for the inactive class as well. This behavior is likely driven by the dataset's class imbalance — most nodes are active, so homophily becomes a statistically advantageous shortcut for the model.

Far from being flawed, the explanation is insightful–it does not mimic the ground truth, but rather points to the root cause of why the inactive class is often misclassified with a low precision of $0.68$: the GNN has failed to capture the intended heterophilous behavior.

### 4.7 Human Evaluation: User Survey

In this work, we advocate a shift from conventional subgraph-based visualizations to natural language explanations. Our motivation stems from the hypothesis that presenting dense graph neighborhoods with high-dimensional node attributes in full detail can overwhelm human cognitive capacity. To test this, we conducted a user study with 60 participants, each responding to 5 A/B test questions, resulting in 300 total comparisons. Each question presented the same explanation for a model prediction in both textual and subgraph form, and participants were asked to indicate their preferred modality. See App. B for survey design details and example screenshots.

We analyzed the results using the *Binomial test* [37] and *McNemar's test* [27]. The Binomial test, applied to the aggregate responses ($200/300$), evaluates whether one modality was preferred significantly more often overall. This yielded a highly significant result ($p$-value $< 0.0001$), indicating that text-based explanations were favored by participants at the population level. Table 3 presents the per-question binomial test results, showing that this preference was consistent and statistically significant in all except Q3. While Q3 does not reach statistical significance, this was by design: we deliberately selected an explanation where the subgraph visualization was small and simple. The goal was to verify that participants were not blindly preferring the text modality due to any prior bias.

To assess within-participant consistency, we additionally conducted McNemar's test using the 'exact Binomial' method. Out of 60 participants, 45 chose text more often, while only 15 preferred subgraph more often. This difference was also statistically significant ($p$-value $= 0.0001$), confirming that the preference for text was not only strong but consistent across individuals.

## 5 Conclusions, Limitations and Future Works

In this work, we presented GNNXEMPLAR, a novel framework for global explanation of node classification in GNNs. Unlike prior explainers that rely on brittle motif discovery and struggle with scale, GNNXEMPLAR explains through exemplar nodes, which serve as semantic anchors for model behavior. To generate human-understandable explanations, we leveraged LLMs to distill the defining characteristics of each exemplar and its neighborhood into concise natural language rules. Our empirical evaluations demonstrate that GNNXEMPLAR outperforms existing methods in fidelity, scalability, and user interpretability, with a user study confirming strong preference for textual explanations.

**Limitations and Future Works:** GNNXEMPLAR operates in a setting with access only to the GNN's embedding space, limiting visibility into how feature-topology interactions influence internal activations. As a result, fine-grained mechanistic understanding of the model's decision process remains out of reach. We plan to study this aspect through the lens of mechanistic interpretability.

## Acknowledgements

The collaborative research with IIT Delhi was supported by Fujitsu Research of India. Burouj Armgaan also acknowledges support from Prime Minister's Research fellowship during this research and the generous travel grant received from ACM/IARCS to support his participation in NeurIPS 2025.

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

# A   Appendix

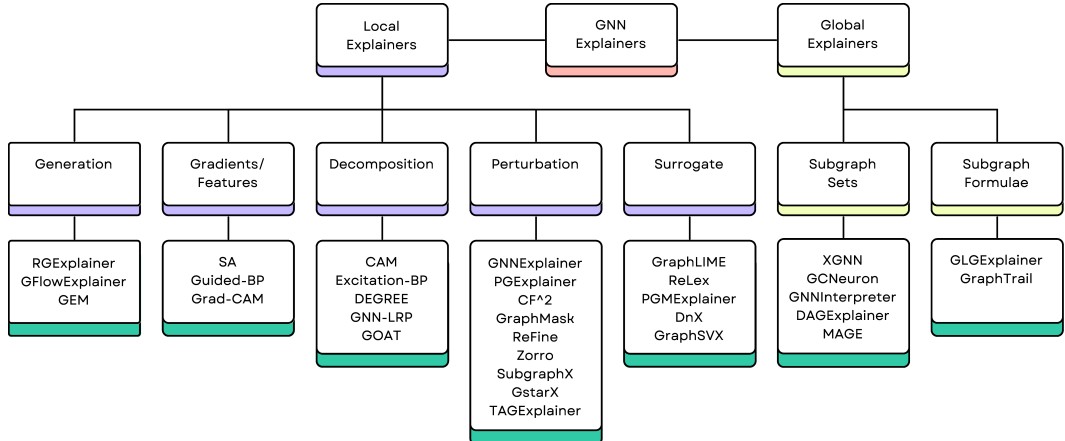

Figure 4: Taxonomy of factual GNN explainers. **Generation:** RGExplainer [36], GFlowExplainer [19], GEM [20]; **Gradient:** SA [3], Guided-BP [3], Grad-CAM [33]; **Decomposition:** CAM [33], Excitation-BP [33], DEGREE [8], GNN-LRP [35], GOAT [23]; **Perturbation:** GNNExplainer [48], PGExplainer [24], CF$^2$ [38], GraphMask [34], ReFine [41], Zorro [9], SubgraphX [52], GstarX [53], TAGExplainer [43]; **Surrogate:** GraphLime [15], ReLex [54], PGM-Explainer [40], DnX [31], GraphSVX [7]; **Subgraph sets:** XGNN [51], GCNeuron [46], GNNInterpreter [42], DAGExplainer [25], MAGE [50]; **Subgraph formulae:** GLGExplainer [2], GraphTrail [1].

## A.1   Experimental setup

**Hardware details**    All experiments were performed on Intel(R) Xeon(R) Gold 6426Y: 64 cores, 126 GB RAM, 2 NVIDIA-L40S GPUs, 45 GiB each running on Ubuntu 20.04.6 LTS.

**Hyper-parameters**    We provide the hyper-parameter values we choose for current pipeline. In sec. 3.2 we fist picked $k$ **for $k$-NN and Reverse $k$-NN**, for all the datasets in experiment we choose $k = 5$. Next in sec. 3.3 for coverage maximization instead of selecting number of points we set the **stopping criterion based on coverage of points**, when the coverage becomes $\geq 95\%$. Then for sec. 3.4 we used **'gemini-1.5-flash' LLM** from google's generativeai package. Then for sampling positive and negative points which will be used in self-refinement we set the **sample size** $= 50$ **each**. We split these samples in **training and validation set in the ratio of** $6 : 4$ respectively. Finally for **stopping criteria of self-refinement** process we stop the iteration when either the **accuracy** $\geq 95$ or if the number of **iterations reaches** 5.

## A.2   Training Details of GNNs

We first write the details about the GNNs we use for all the datasets,

- For TAGCora, we pick the GAT model (with 2 layers and 64 hidden dimension) from paper [21] where they officially use this dataset.
- For WikiCS dataset we pick the official implementation of GCN used in their paper [29]. And use 2 layer model with a hidden dimension of 64.
- For ogbn-arxiv, we pick the official GCN from *ogb repository* for node classification on arxiv, we used a 3 layered version of it with hidden dimension as 256.
- For Citeseer, amazon-ratings, questions and minesweeper we use the same model as use in ogbn-arxiv dataset and for all these datasets we use 3 layers with hidden dimension of 64.
- For BA-Shapes, we use the model use in the paper [48] with 3 layers, a hidden dimension of 20.

Since none of the above implementations provide trained model weights so we train the GNNs on these datasets from scratch. For training all the GNNs we use the standard training pipeline of pytorch which resembles closely to what all the above models use in their implementation. We train all the models for 250 epochs and choose the learning rates from the set $\{0.005, 0.01\}$ with Adam

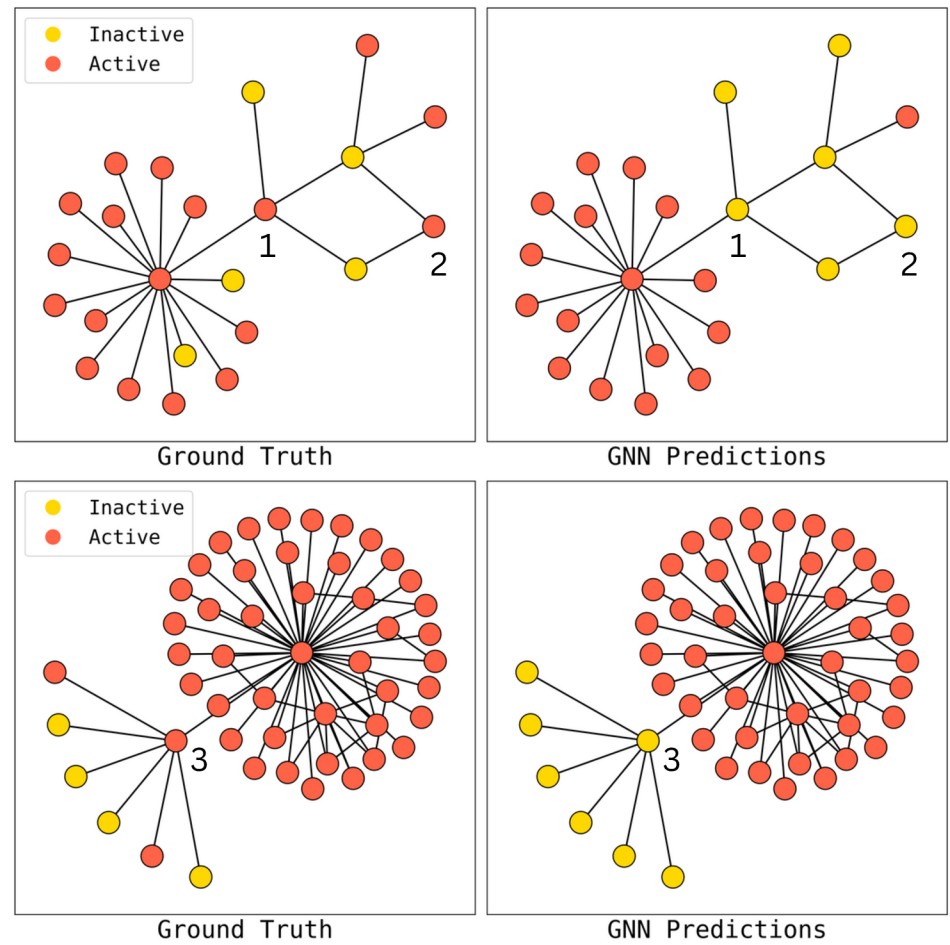

Figure 5: Subgraphs from the Questions dataset. Notice how active nodes 1, 2, and 3 surrounded by inactive nodes are predicted as inactive. This misclassification pattern is the signal that GNNXEM-PLAR picks up when learning the signature for the inactive class.

optimizer and a weight decay of $5e - 4$. We report the best accuracies of trained GNNs in table 4. Although we try to get as close as possible to the quoted numbers in the datasets official papers there could still be some deviation because of random initialization of model weights.

Table 4: GNN performance

| Homophilous | | Heterophilous | |
|---|---|---|---|
| **Name** | **Accuracy** | **Name** | **Accuracy** |
| TAGCora | 73.84 | Amazon-ratings | 46.43 |
| Citeseer | 66.20 | Minesweeper | 80.36 |
| WikiCS | 78.40 | Questions | 97.07 |
| ogbn-arxiv | 62.58 | BA-Shapes | 95.71 |

## A.3 Proof of Lemma 1

Let $\mathbb{S} \subseteq \mathcal{V}_{tr}$ be a set of $z$ nodes sampled uniformly at random from the full training node set $\mathcal{V}_{tr}$. For a fixed node $v \in \mathcal{V}_{tr}$, define the indicator random variable $X_u^v$ for each $u \in \mathbb{S}$ such that:

$$X_u^v = \begin{cases} 1 & \text{if } v \in k\text{-NN}(u) \\ 0 & \text{otherwise} \end{cases} \tag{3}$$

The total number of times $v$ appears in the $k$-NN lists of sampled nodes is given by:

$$X^v = \sum_{u \in \mathbb{S}} X_u^v = z \cdot \widetilde{\Pi}(v) \tag{4}$$

Since nodes are sampled independently and uniformly, we have $E[X^v] = z \cdot \Pi(v)$, where $\Pi(v)$ is the true representative power of node $v$. Thus, $X^v$ follows a Binomial distribution: $X^v \sim \text{Binomial}(z, \Pi(v))$.

Applying Chernoff bounds for any $\epsilon \geq 0$:

$$\text{Upper tail:} \quad P\left(X^v \geq (1 + \epsilon) z \Pi(v)\right) \leq \exp\left(-\frac{\epsilon^2}{2 + \epsilon} z \Pi(v)\right) \tag{5}$$

$$\text{Lower tail:} \quad P\left(X^v \leq (1 - \epsilon) z \Pi(v)\right) \leq \exp\left(-\frac{\epsilon^2}{2} z \Pi(v)\right) \tag{6}$$

Combining both bounds:

$$P\left(|X^v - z\Pi(v)| \geq \epsilon z \Pi(v)\right) \leq 2 \exp\left(-\frac{\epsilon^2}{2 + \epsilon} z \Pi(v)\right) \tag{7}$$

$$\Rightarrow P\left(\left|z\widetilde{\Pi}(v) - z\Pi(v)\right| \geq \epsilon z \Pi(v)\right) \leq 2 \exp\left(-\frac{\epsilon^2}{2 + \epsilon} z \Pi(v)\right) \tag{8}$$

$$\Rightarrow P\left(\left|\widetilde{\Pi}(v) - \Pi(v)\right| \geq \epsilon \Pi(v)\right) \leq 2 \exp\left(-\frac{\epsilon^2}{2 + \epsilon} z \Pi(v)\right) \tag{9}$$

Substituting $\theta = \epsilon \Pi(v)$ from Lemma 1 in Eq. 9, we rewrite Eq. 9 as:

$$P\left(\left|\widetilde{\Pi}(v) - \Pi(v)\right| \geq \theta\right) \leq 2 \exp\left(-\frac{\left(\frac{\theta}{\Pi(v)}\right)^2}{2 + \frac{\theta}{\Pi(v)}} z \Pi(v)\right) \tag{10}$$

$$\leq 2 \exp\left(-\frac{\theta^2}{(2\Pi(v) + \theta)} z\right) \tag{11}$$

Since $\Pi(v) \leq 1$, we obtain:

$$P\left(\left|\widetilde{\Pi}(v) - \Pi(v)\right| \geq \theta\right) \leq 2 \exp\left(-\frac{\theta^2}{2 + \theta} z\right) \tag{12}$$

To ensure this probability is at most $\delta$, we solve:

$$2 \exp\left(-\frac{\theta^2}{2 + \theta} z\right) \leq \delta \tag{13}$$

$$\Rightarrow \ln\left(\frac{2}{\delta}\right) \leq \frac{\theta^2}{2 + \theta} z \tag{14}$$

$$\Rightarrow z \geq \frac{\ln\left(\frac{2}{\delta}\right)(2 + \theta)}{\theta^2} \tag{15}$$

Hence, if we sample at least $z \geq \frac{\ln\left(\frac{2}{\delta}\right)(2 + \theta)}{\theta^2}$ nodes in $\mathbb{S}$, then for any $v \in \mathcal{V}$:

$$\widetilde{\Pi}(v) \in [\Pi(v) - \theta, \ \Pi(v) + \theta] \quad \text{with probability at least } 1 - \delta. \qquad \square \tag{16}$$

## A.4 NP-hardness: Proof of Theorem 1

PROOF. We prove NP-hardness of the representative node selection problem by a reduction from the classical *Set Cover* problem [5].

**Definition 7** (Set Cover). *Given a budget $b$ and a collection of subsets $\mathcal{S} = \{S_1, \ldots, S_m\}$ of a universe $U = \{u_1, \ldots, u_n\}$ such that $S_i \subseteq U$ for all $i$, the Set Cover problem asks whether there exists a subcollection $\mathcal{S}' \subseteq \mathcal{S}$ of size $|\mathcal{S}'| = b$ such that $\bigcup_{S_i \in \mathcal{S}'} S_i = U$.*

Given an instance of Set Cover $\langle \mathcal{S}, U \rangle, b$, we construct a graph $G = (V, E)$ and corresponding node embeddings such that solving our representative selection problem on this graph is equivalent to solving the Set Cover instance.

We create a training node set $\mathcal{V}_{tr} = \mathcal{V}_U \cup \mathcal{V}_\mathcal{S}$, where:

- For each element $u_j \in U$, we create a node $v_{u_j} \in \mathcal{V}_U$.
- For each subset $S_i \in \mathcal{S}$, we create a node $v_{S_i} \in \mathcal{V}_\mathcal{S}$.

We define the $k$-NN relationship over the learned GNN embeddings such that for each $u_j \in S_i$, node $v_{S_i} \in k\text{-NN}(v_{u_j})$, i.e., $v_{u_j} \in Rev-\mathrm{k}-NN(v_{S_i})$. This means that $v_{S_i}$ *represents* all $v_{u_j}$ for which $u_j \in S_i$.

Now, consider the exemplar selection problem: choose a set $\mathbb{A} \subseteq \mathcal{V}$ of size $b$ that maximizes the representative power:

$$\Pi(\mathbb{A}) = \frac{|\{v \in \mathcal{V}_{tr} \mid \exists v' \in \mathbb{A} \text{ such that } v \in \text{Rev-}k\text{-NN}(v')\}|}{|\mathcal{V}_{tr}|} \tag{17}$$

Our construction ensures that $\Pi(\mathbb{A}) = \frac{|\mathcal{V}_U|}{|\mathcal{V}_{tr}|}$ if and only if all nodes in $\mathcal{V}_U$ are covered by the reverse $k$-NNs of nodes in $\mathbb{A}$. This is equivalent to finding $b$ nodes from $\mathcal{V}_\mathcal{S}$ (i.e., subsets in the original Set Cover instance) whose reverse $k$-NNs jointly cover all nodes in $\mathcal{V}_U$ (i.e., elements in $U$). Thus, there exists a set cover of size $b$ if and only if there exists a node subset $\mathbb{A} \subseteq \mathcal{V}_{tr}$ of size $b$ such that $\Pi(\mathbb{A}) = \frac{|\mathcal{V}_U|}{|\mathcal{V}_{tr}|}$. $\qquad\square$

## A.5 Proofs of Monotonicity and Submodularity

**Theorem 3** (Monotonicity). *Let $\mathbb{A} \subseteq \mathbb{A}'$ be two subsets of nodes in the graph. Then,*

$$\Pi(\mathbb{A}') - \Pi(\mathbb{A}) \geq 0$$

*Proof.* Recall that $\Pi(\mathbb{A}) = \frac{|\bigcup_{v \in \mathbb{A}} \text{Rev-}k\text{-NN}(v)|}{|\mathcal{V}_{tr}|}$. The denominator $|\mathcal{V}_{tr}|$ is constant, so we only need to show that the numerator is monotonic.

Since $\mathbb{A}' \supseteq \mathbb{A}$, we have:

$$\bigcup_{v \in \mathbb{A}'} \text{Rev-}k\text{-NN}(v) \supseteq \bigcup_{v \in \mathbb{A}} \text{Rev-}k\text{-NN}(v)$$

because the union operator is monotonic under set inclusion. Therefore, the size of the covered set cannot decrease as $\mathbb{A}$ grows. This implies:

$$\Pi(\mathbb{A}') \geq \Pi(\mathbb{A})$$

$\qquad\square$

**Theorem 4** (Submodularity). *Let $\mathbb{A} \subseteq \mathbb{A}'$ and $v$ be any node in the graph not already in $\mathbb{A}'$. Then,*

$$\Pi(\mathbb{A}' \cup \{v\}) - \Pi(\mathbb{A}') \leq \Pi(\mathbb{A} \cup \{v\}) - \Pi(\mathbb{A})$$

PROOF BY CONTRADICTION. Assume instead that:

$$\Pi(\mathbb{A}' \cup \{v\}) - \Pi(\mathbb{A}') > \Pi(\mathbb{A} \cup \{v\}) - \Pi(\mathbb{A}) \tag{18}$$

Let $R_v = \text{Rev-}k\text{-NN}(v)$ denote the set of nodes that consider $v$ among their $k$ nearest neighbors. The marginal gain of adding $v$ to a set $\mathbb{A}$ is:

$$\left| R_v \setminus \bigcup_{v' \in \mathbb{A}} \text{Rev-}k\text{-NN}(v') \right|$$

Inequality (18) implies:

$$\left|R_v \setminus \bigcup_{v' \in \mathbb{A}'} \text{Rev-}k\text{-NN}(v')\right| > \left|R_v \setminus \bigcup_{v' \in \mathbb{A}} \text{Rev-}k\text{-NN}(v')\right|$$

But since $\mathbb{A}' \supseteq \mathbb{A}$, we have:

$$\bigcup_{v' \in \mathbb{A}} \text{Rev-}k\text{-NN}(v') \subseteq \bigcup_{v' \in \mathbb{A}'} \text{Rev-}k\text{-NN}(v')$$

which implies the *reverse* inequality:

$$\left|R_v \setminus \bigcup_{v' \in \mathbb{A}'} \text{Rev-}k\text{-NN}(v')\right| \leq \left|R_v \setminus \bigcup_{v' \in \mathbb{A}} \text{Rev-}k\text{-NN}(v')\right|$$

This contradicts our assumption in Eq. (18), completing the proof. $\square$

### A.6 Proof of Monotonicity and Submodularity under Estimated Rev-k-NN

Let $\widetilde{\text{Rev-}k\text{-NN}}(v)$ denote the approximate reverse k-NN set of a node $v$, and $\widetilde{\Pi}(A)$ denote the approximate coverage function computed using these sets. The greedy algorithm in Theorem 2 provides a $(1 - 1/e)$-approximation to the maximization of $\widetilde{\Pi}(A)$, since the proof relies solely on submodularity and monotonicity of the input function—both of which are preserved under the approximation (as the union of approximate sets still yields a valid set function).

To relate this back to the true optimum $\Pi(A^*)$, we leverage the uniform error bound from Lemma 1 (Appendix A.3), which ensures that for any set $A$:

$$|\widetilde{\Pi}(A) - \Pi(A)| \leq \theta \quad \text{with high probability}(\geq 1 - \delta).$$

Let $\widetilde{A}$ be the set returned by greedy maximization over $\widetilde{\Pi}$, and $A^*$ be the optimal set under the true objective $\Pi$. The greedy guarantee gives:

$$\widetilde{\Pi}(\widetilde{A}) \geq \left(1 - \frac{1}{e}\right) \widetilde{\Pi}(A^*).$$

By the uniform bound, we know $\widetilde{\Pi}(A^*) \geq \Pi(A^*) - \theta$, so:

$$\widetilde{\Pi}(\widetilde{A}) \geq \left(1 - \frac{1}{e}\right)(\Pi(A^*) - \theta) = \left(1 - \frac{1}{e}\right)\Pi(A^*) - \left(1 - \frac{1}{e}\right)\theta.$$

Thus, even when using approximate Rev-k-NN sets, the greedy algorithm achieves the following guarantee on estimated coverage:

$$\boxed{\widetilde{\Pi}(\widetilde{A}) \geq \left(1 - \frac{1}{e}\right)\Pi(A^*) - \left(1 - \frac{1}{e}\right)\theta}$$

$\square$

### A.7 Impact of parameters

All experiments in this section are conducted on two large-scale benchmarks—WikiCS (homophilous) and Questions (heterophilous)—to evaluate three critical hyperparameters:

**Rev-$k$-NN neighbourhood size ($k$).** Fig. 17 shows that a moderate choice of $k$ maximizes fidelity for both WikiCS and Questions. On WikiCS, increasing $k$ from 1 to 5 raises mean fidelity from 0.72 to 0.78, after which fidelity declines (0.72 at 20, 0.69 at 100). Likewise on Questions, fidelity peaks at 0.92 for $k = 5$ and then falls to 0.86 at $k = 100$. These trends indicate that too small a prototype set under-covers the class distribution, while too large a set introduces noisy, less representative examples.

Table 5: Accuracy of the underlying GNN and fidelity of GNNXEMPLAR across different GNN architectures on TAGCora

| Architecture | GNN Accuracy (%) | Fidelity (%) |
|---|---|---|
| GraphSAGE | 68.91 | 76.75 |
| GCN | 68.71 | 79.11 |
| GAT | 73.84 | 83.00 |

**Training set sampling size.** Fig. 18 evaluates the number of positive and negative nodes used to generate feedback in each self-refinement iteration. Fidelity on both datasets peaks at a sampling size of 20 (0.78 for WikiCS, 0.92 for Questions). Smaller sample sizes (1 or 5) under-represent mis-classified patterns, yielding lower fidelity, whereas excessively large samples (100 or 200) introduce so much data that the LLM struggles to distill the most salient corrective signals

**Sampling rate for exemplar selection.** Fig. 19 shows fidelity and runtime as a function of the fraction of nodes used to compute Rev-$k$-NN prototypes. Fidelity rapidly saturates by 5–10 % sampling—achieving 0.75–0.78 on WikiCS and 0.86–0.92 on Questions—while full sampling (100 %) incurs orders-of-magnitude higher runtimes (0.7 s $\to$ 690 s on WikiCS; 0.01 s $\to$ 41.6 s on Questions) without significant fidelity gains. This demonstrates that a small fraction of the data suffices to produce explanations of near–state-of-the-art fidelity at minimal computational cost.

**Practical recommendations.** In light of these results, we recommend setting $k = 5$, using a positive–negative sample size of 20 per iteration, and sampling 5–10 % of nodes for exemplar selection to balance explanation quality and efficiency.

### A.8   Robustness Across GNN Architectures

To assess the generalizability to GNN architectures, we apply it to three widely-used GNN back-bones—GraphSAGE, GCN, and GAT—each trained on the same node-classification task for the TAGCora dataset. Table 5 summarizes both the GNN's accuracy and the fidelity of our extracted rules.

Despite differences in representational power and message-passing mechanisms, GNNXEMPLAR achieves consistently high fidelity ranging from 76.8 % for GraphSAGE to 83.0 % for GAT. These results confirm that GNNXEMPLAR reliably translates diverse learned graph representations into interpretable symbolic rules.

### A.9   Dataset Descriptions

We evaluate our method on eight established node-classification benchmarks, chosen to span homophilous and heterophilous graphs, text-driven and purely structural domains:

- **TAGCora** A citation network of 2,708 computer-science papers connected by 10,556 directed edges. Each node corresponds to a paper and is featurized by its title and abstract text. The task is to predict one of seven topic categories at the node level.
- **CiteSeer** A citation network of 3312 publications partitioned into six topic labels. Edges indicate citation links. Each paper is represented by a 3703-dimensional binary bag-of-words vector.
- **ogbn-arxiv** A directed graph of all CS arXiv papers indexed by MAG, with 169343 nodes and 1166243 citation edges. Node features are 128-dimensional vectors formed by averaging skip-gram embeddings over title and abstract text. The 40 subject-area labels (e.g. cs.AI, cs.LG) yield a 40-way classification task.
- **WikiCS** A homogeneous hyperlink network of 11701 Wikipedia articles across ten computer-science categories, connected by 216123 edges. Node features are 300-dimensional averages of pretrained GloVe word embeddings over the article text.

- **Amazon-ratings** A product co-purchase graph (24492 nodes, 93050 edges) where edges link frequently co-bought items. The goal is to predict discretized average review scores (five classes). Each product is featurized by the mean of fastText embeddings over its description.
- **Minesweeper** A synthetic $100 \times 100$ grid (10000 nodes, $\approx$39402 edges) modeling the Minesweeper game. Twenty percent of cells contain mines. Node features are one-hot encodings of the true count of adjacent mines, with a binary "unknown" flag on 50% of nodes.
- **Questions** A user-interaction graph from Yandex Q (48921 nodes, 153540 edges), where an edge indicates one user answered another's question over one year. The binary classification task predicts which users remain active. Node features combine fastText averages over user-profile text with a binary "no profile" indicator.
- **BA-Shapes** A synthetic heterophilous graph built by attaching 80 five-node "house" motifs to a 300-node Barabási–Albert backbone, then adding random edges (10% of number of nodes). Each node is labeled as *tip*, *middle*, *base*, or *background*.

# B  Human Evaluation

We designed and executed a survey in the form of an **A/B test** to compare the interpretability of text-based explanations versus traditional subgraph-based explanations.

**Survey Setup**  Participants were shown global explanations for the same GNN model, one in the form of a subgraph(s) and the other as a textual rule. Their task was to evaluate both independently and indicate which modality better communicated the model's reasoning for classifying a particular class of nodes. We explicitly clarified the meaning of *global explanations* — those meant to summarize common decision patterns across all nodes of a given class — in contrast to *local explanations* that are tailored to individual nodes. To help participants make an informed judgment, we included clear and detailed instructions at the beginning of the survey. These instructions emphasized that:

- Each explanation should be evaluated in isolation.
- Participants should not attempt to align the textual and subgraph formats.
- The goal is to judge which modality stands better on its own in helping them understand the GNN's reasoning.

**Participant Demographics**  We recruited 60 participants with at least a basic proficiency in machine learning for 5 A/B tests amounting for 300 total comparisons. All respondents came from a computer science or engineering background, and we deliberately sampled both experts, i.e., researchers actively working on graphs, and non-experts, who may work in other ML domains but are still familiar with model reasoning and classification tasks. This balance allowed us to assess interpretability across varying levels of familiarity with graph-structured data.

**Statistical Tests**  To evaluate user preferences between the two explanation modalities presented across five A/B questions, we conducted both a Binomial test and McNemar's test. The **Binomial test**, applied to the aggregate responses (275 total comparisons), assesses whether one modality was preferred significantly more often overall. While informative, this test treats each question response as independent and does not account for within-subject consistency. To address this, we additionally used **McNemar's test**, which considers per-user paired choices across the five questions, capturing whether individuals systematically favored one modality over the other. Together, these tests provide complementary insights: the binomial test reflects population-level preference strength, while McNemar's test confirms the consistency of that preference at the individual level.

**Survey Materials and Transparency**  To ensure full transparency and reproducibility, we include screenshots of key survey components in the Appendix. Fig. 6 shows the welcome page, which introduced the task, explained the distinction between global and local explanations, and outlined the survey objective. Fig. 7 presents the instructional page detailing how to interpret and evaluate the two explanation modalities. Fig. 8 shows a representative A/B test question featuring a dataset description and both types of explanations. Finally, Fig. 9 illustrates the evaluation criteria provided before each A/B comparison to guide participants in making modality-specific judgments. These materials demonstrate the care taken to ensure participants understood the task and the evaluation criteria, lending credibility to the findings reported in Sec. 4.7.

**Controlling for Content Differences Across Modalities**  The goal of this survey was to compare the modalities of explanation—text versus subgraph—not their content richness. Since our textual explanations are generated automatically, we needed to ensure that the subgraph explanations were as close in content as possible to the text ones, so that the only difference being judged by participants was the modality, not the information conveyed.

Our textual explanations include both structural and feature-based insights. For example, Fig. 14 shows the explanation for the Rule Learning class in TAGCora: *Publications in "Rule Learning" contain keywords such as "inductive logic programming," "meta-knowledge," or "hypothesis space," and/or their immediate and 2-hop citation neighborhoods are predominantly Rule Learning papers.*

This explanation draws on both graph topology and node features (keywords). However, subgraph visualizations cannot communicate high-dimensional features — how would one visually encode a 128-dimensional feature vector within a graph view?

To address this mismatch and ensure a fair comparison, we intentionally disabled feature information while generating the text explanations used in the survey. This ensured that both explanation types were grounded purely in structure, making the comparison about modality alone. Without this control, the text-based explanations would have had an inherent informational advantage, introducing bias into the survey.

This content difference is illustrated in Fig. 14 and 8. The former shows a feature-rich explanation, while the latter corresponds to the feature-neutral version used in the survey.

Table 6: Precision

|  | Homophilous | | | | Heterophilous | | | |
|---|---|---|---|---|---|---|---|---|
|  | **TAGCora** | **Citeseer** | **WikiCS** | **ogbn-arxiv** | **Amazon-ratings** | **Questions** | **Minesweeper** | **BA-Shapes** |
| **GNNInterpreter** | NA | $0.50 \pm 0.00$ | NA | NA | NA | NA | $0.50 \pm 0.00$ | $0.54 \pm 0.10$ |
| **GCNeuron** | $0.32 \pm 0.00$ | $0.16 \pm 0.00$ | NA | NA | $0.75 \pm 0.10$ | NA | $0.58 \pm 0.00$ | $0.16 \pm 0.00$ |
| **GLGExplainer** | NF | NF | OOM | OOM | NF | OOM | $0.25 \pm 0.01$ | $0.37 \pm 0.09$ |
| **Ours** | $\mathbf{0.85 \pm 0.03}$ | $\mathbf{0.95 \pm 0.05}$ | $\mathbf{0.98 \pm 0.00}$ | $\mathbf{0.99 \pm 0.00}$ | $\mathbf{0.81 \pm 0.05}$ | $\mathbf{0.96 \pm 0.00}$ | $\mathbf{0.81 \pm 0.02}$ | $\mathbf{0.95 \pm 0.01}$ |

Table 7: Recall

|  | Homophilous | | | | Heterophilous | | | |
|---|---|---|---|---|---|---|---|---|
|  | **TAGCora** | **Citeseer** | **WikiCS** | **ogbn-arxiv** | **Amazon-ratings** | **Questions** | **Minesweeper** | **BA-Shapes** |
| **GNNInterpreter** | NA | $1.00 \pm 0.00$ | NA | NA | NA | NA | $1.00 \pm 0.00$ | $0.29 \pm 0.10$ |
| **GCNeuron** | $0.17 \pm 0.00$ | $0.33 \pm 0.00$ | NA | NA | $0.27 \pm 0.00$ | NA | $0.53 \pm 0.00$ | $0.33 \pm 0.00$ |
| **GLGExplainer** | NF | NF | OOM | OOM | NF | OOM | $0.49 \pm 0.01$ | $0.22 \pm 0.12$ |
| **Ours** | $\mathbf{0.82 \pm 0.02}$ | $\mathbf{0.89 \pm 0.06}$ | $\mathbf{0.58 \pm 0.03}$ | $\mathbf{0.67 \pm 0.02}$ | $\mathbf{0.80 \pm 0.02}$ | $\mathbf{0.81 \pm 0.04}$ | $\mathbf{0.96 \pm 0.00}$ | $\mathbf{0.91 \pm 0.00}$ |

Table 8: F1 Score

|  | Homophilous | | | | Heterophilous | | | |
|---|---|---|---|---|---|---|---|---|
|  | **TAGCora** | **Citeseer** | **WikiCS** | **ogbn-arxiv** | **Amazon-ratings** | **Questions** | **Minesweeper** | **BA-Shapes** |
| **GNNInterpreter** | NA | $0.67 \pm 0.00$ | NA | NA | NA | NA | $0.67 \pm 0.00$ | $0.34 \pm 0.10$ |
| **GCNeuron** | $0.15 \pm 0.00$ | $0.22 \pm 0.00$ | NA | NA | $0.26 \pm 0.00$ | NA | $0.45 \pm 0.00$ | $0.22 \pm 0.00$ |
| **GLGExplainer** | NF | NF | OOM | OOM | NF | OOM | $0.33 \pm 0.01$ | $0.24 \pm 0.11$ |
| **Ours** | $\mathbf{0.82 \pm 0.01}$ | $\mathbf{0.92 \pm 0.03}$ | $\mathbf{0.71 \pm 0.03}$ | $\mathbf{0.77 \pm 0.01}$ | $\mathbf{0.80 \pm 0.02}$ | $\mathbf{0.86 \pm 0.03}$ | $\mathbf{0.87 \pm 0.01}$ | $\mathbf{0.93 \pm 0.00}$ |

## Comparing global explanation modalities for Graph Neural Networks

👋 Thank you for participating in this study on the explainability of Graph Neural Networks (GNNs)!

In this study, you will explore how a trained GNN classifies nodes into particular classes. For each classification task, you will be presented with:
• Two global explanations of the same GNN
• One in the form of a subgraph visualization
• One as a textual description

🔍 **What are Explanations?**
An explanation tells you why a model makes a particular prediction — what pattern it has picked up. In this survey, explanations take the form of subgraph visualizations and short textual summaries that highlight the patterns the GNN has learned.

🔍 **What are "Global" Explanations?**
The explanations in this survey are meant to be global explanations — this is a key point:
• Unlike local explanations, which justify why *one specific node* was classified a certain way, resulting in one explanation per node,
• Global explanations aim to show the common reasoning the model uses across all nodes of a given class, resulting in one explanation per class of nodes.

✏️ **Your Task**
• Review both explanations independently.
• Decide which modality — subgraph or text — better helps you understand how the model thinks about the entire class.

✅ **A Few Notes**
• There are no right or wrong answers — we're interested in your preference.
• Just let us know which explanation you find more helpful and understandable.

We appreciate your time and insights!

🎯 **Target Audience:** Individuals with an engineering background—preferably in Computer Science—and a foundational understanding of machine learning.
⏳ **Duration:** 7-10 minutes
🔵 PS: **SurveyCircle** users receive points for their participation, which can be used to recruit free survey participants at SurveyCircle.com

Figure 6: Welcome page of the user study. This page introduced participants to the study's objective, explained the distinction between local and global explanations, described the two explanation modalities, and clarified the task expectations. It also included details about the target audience, estimated completion time, and participant incentives.

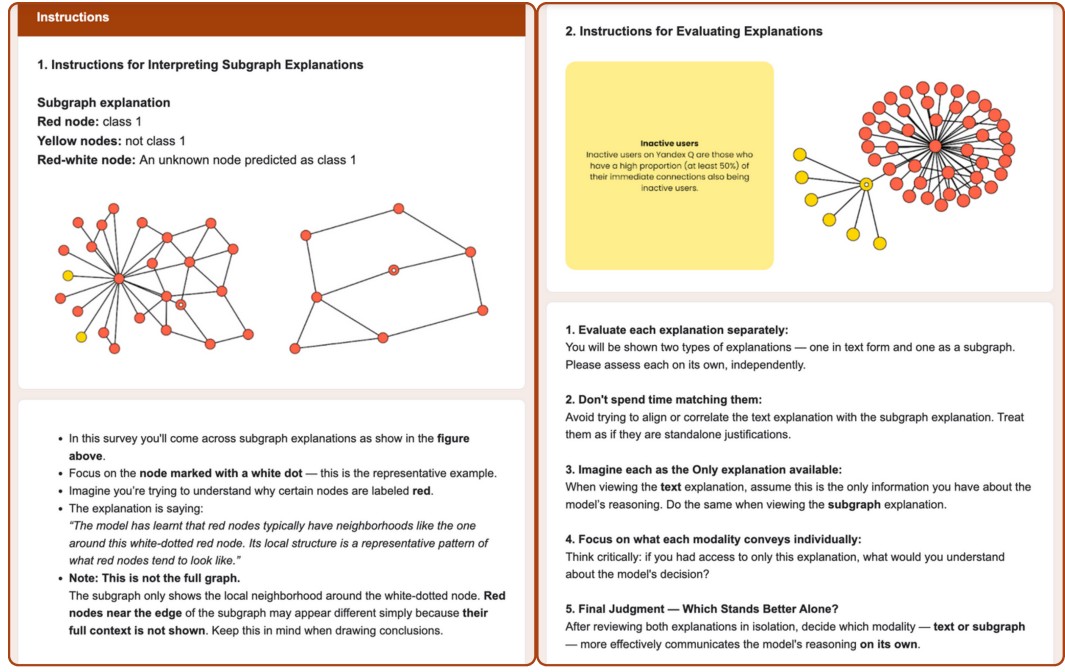

Figure 7: Instructions shown to participants prior to the survey. These guidelines clarified how to interpret subgraph and text-based explanations, emphasized evaluating each modality independently, and explained the concept of global explanations to ensure consistent and meaningful user responses.

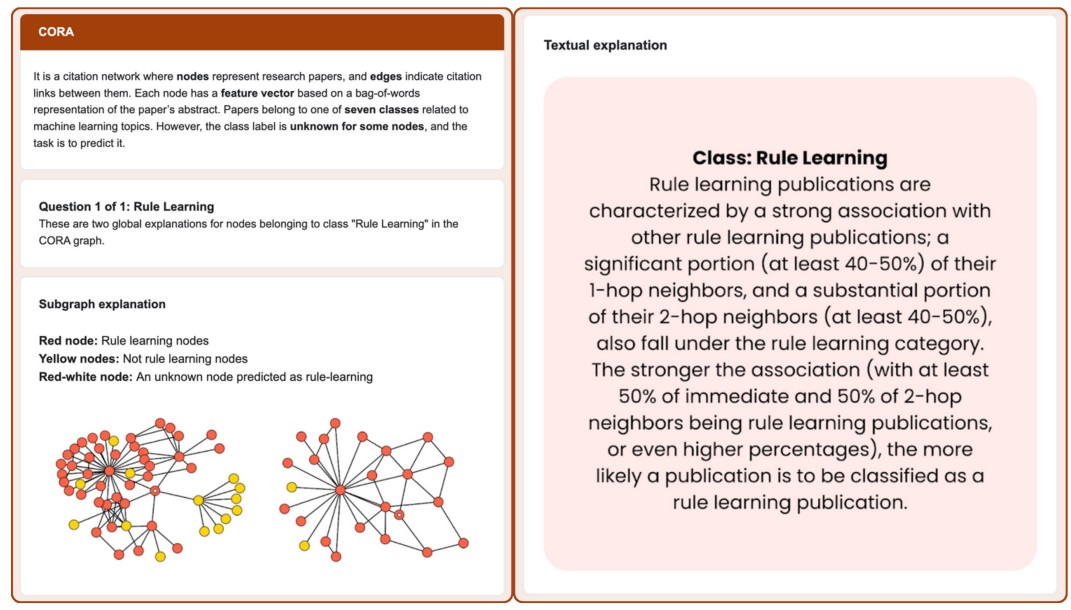

Figure 8: Example survey question shown to participants. This page presents the task context for a global explanation comparison—participants were shown a dataset description followed by two explanation modalities (text-based and subgraph-based) for the same model prediction and asked to select their preferred one.

**Answer 1 of 1: Which global explanation do you prefer?** *

Please consider the following when making your choice:
1. Will the explanation generalize to *all* nodes of the selected class — not just individual examples?

2. Is the modality (graphical or textual) well-suited to how *you* prefer to interpret information?

3. How clearly does the explanation communicate the model's reasoning for classifying nodes into this class?

4. If you had to rely on one explanation to understand or trust the model's behavior, which would you choose?

○ Text

○ Subgraph

Figure 9: Evaluation guidelines provided before each A/B comparison. Participants were instructed to evaluate each explanation independently, avoid aligning the two modalities, and judge which one more effectively communicates the model's reasoning on its own. These instructions ensured unbiased, modality-specific assessments.

```
You are given an exemplar node and a set of nodes that belong to its Rev-k-NN
    set (from a GNN embedding). Also provided are some nodes that do NOT
    belong to this Rev-k-NN set. Please propose an INITIAL symbolic
    rule-based formula that outputs 1 for Rev-k-NN set nodes and 0 for
    non-Rev-k-NN set nodes. Remember that your rules should not be generic
    and should be specific to the exemplar node and its Rev-k-NN set. Rely
    ONLY on node features and local adjacency information. Focus on finding
    what features and structural neighborhood information makes the Rev-k-NN
    set nodes similar and the non-Rev-k-NN set nodes dissimilar to the
    exemplar node. Use them to form the rules.

Output your explanation in JSON with a top-level rules list and an
    interpretation key. Also implement a Python function called
    classify_node() that takes a node_description as input and returns 1 or
    0. Ensure the code can run independently without external file
    dependencies and without any errors. Avoid using regex, or importing any
    more external libraries. Also ensure that all the code is inside the
    classify_node function only, and nothing should be outside it.

Exemplar node: 11757
Exemplar node description:
{
   'node_id': 11757,
   'features': [0.06785757, 0.11838776, 3.94833946],
   '1-hop': {'neighbor_class_freq': {1: 1.0}},
   '2-hop': {'neighbor_class_freq': {1: 1.0}}
}
Rev-k-NN set nodes: [2593, 9375]
Non-Rev-k-NN set nodes: [32414, 14315]

Descriptions of Rev-k-NN set nodes:

Node 2593: {'node_id': 2593, 'features': [0.07221082, 0.11624873,
    3.92639644], '1-hop': {'neighbor_class_freq': {1: 1.0}}, '2-hop':
    {'neighbor_class_freq': {0: 0.4736842, 1: 0.5263158}}}

Node 9375: {'node_id': 9375, 'features': [0.13466489, 0.14399602,
    0.05334831], '1-hop': {'neighbor_class_freq': {1: 1.0}}, '2-hop':
    {'neighbor_class_freq': {0: 0.4736842, 1: 0.5263158}}}

Descriptions of non-Rev-k-NN set nodes:

Node 32414: {'node_id': 32414, 'features': [0.05580222, 0.08147032,
    -0.06319338], '1-hop': {'neighbor_class_freq': {0: 1.0}}, '2-hop':
    {'neighbor_class_freq': {0: 1.0}}}

Node 14315: {'node_id': 14315, 'features': [0.08261909, 0.50772273,
    0.49023183], '1-hop': {'neighbor_class_freq': {0: 1.0}}, '2-hop':
    {'neighbor_class_freq': {0: 1.0}}}
```

Figure 10: Query prompt given to the LLM on the questions dataset

```
Below is the CURRENT formula:
def classify_node(node_description):
    try:
        if node_description['features'][2] > 3.0 and
        node_description['1-hop']['neighbor_class_freq'].get(1, 0) >= 0.5:
            return 1
        else:
            return 0
    except (KeyError, IndexError):
        return 0

It was evaluated on Rev-k-NN set nodes (should be 1) and non-Rev-k-NN set
nodes (should be 0). It yielded 2 false negatives and 0 false positives.

False Negatives (Rev-k-NN set nodes misclassified as non-Rev-k-NN set):
Node 1920: {'node_id': 1920, 'features': array([0.04922352, 0.19213917,
0.6143764 ]),
            '1-hop': {'neighbor_class_freq': {0: 0.5714, 1: 0.4286}},
            '2-hop': {'neighbor_class_freq': {0: 0.9715, 1: 0.0285}}}
Node 15365: {'node_id': 15365, 'features': array([0.0492574 , 0.19516333,
0.59393265]),
             '1-hop': {'neighbor_class_freq': {1: 1.0}},
             '2-hop': {'neighbor_class_freq': {0: 0.25, 1: 0.75}}}

False Positives (non-Rev-k-NN set nodes misclassified as Rev-k-NN set):
None.

Your goal is to improve the formula so that it does not make these errors
while preserving its correct predictions. Provide detailed feedback on which
condition to add, remove, or modify, and state the single most important
change to make. Do not supply the revised code yet.
```

Figure 11: Feedback prompt given to the LLM on the questions dataset

```
Below is the history of previous iterations:
Iteration 0:
Formula:

def classify_node(node_description):
    try:
        if node_description['features'][2] > 3.0 and
        node_description['1-hop']['neighbor_class_freq'].get(1, 0) >= 0.5:
            return 1
        else:
            return 0
    except (KeyError, IndexError):
        return 0
```

Positives Accuracy: 0.46031746031746035, Negatives Accuracy: 1.0
Actionable Feedback:
The single most important change to improve the formula is to increase the
threshold for node['1-hop']['neighbor_class_freq'].get(1, 0) from 0.5 to 0.7.
The current threshold is too lenient, allowing nodes with a significant
proportion of non-class 1 neighbors to be incorrectly classified as Rev-k-NN
set members. Raising the threshold will better discriminate between Rev-k-NN
set and non-Rev-k-NN set nodes based on 1-hop neighborhood information.
Further adjustments to the feature threshold might also be beneficial, but
addressing the neighbor frequency threshold is the most impactful initial
step.

Using the history above, and the last feedback, please produce a REVISED
formula in JSON format with a top-level 'rules' list and an 'interpretation'
key. Your goal is to improve the accuracy metrics by addressing the feedback
provided. You must achieve at least 95% positive and negative accuracy. Also,
implement a Python function called classify_node() that takes a
node_description as input and returns 1 or 0. Ensure the code can run
independently without external file dependencies and without any errors.
Avoid using regex, or importing any more external libraries. Also ensure that
all the code is inside the classify_node function only, and nothing should be
outside it.

Figure 12: Refine prompt given to the LLM on the questions dataset

```python
def classify_class_0(node_description):
    features = node_description.get('features', '')
    one_hop  = node_description.get('1-hop',  {}).get('
        neighbor_class_freq', {})
    two_hop  = node_description.get('2-hop',  {}).get('
        neighbor_class_freq', {})

    # Exemplar #1 signature
    score1  = 0.5 * int(any(k in features
                           for k in ['ILP','meta-knowledge','
                               hypothesis space']))
    score1 += 0.3 * int(one_hop.get(0,0) > 0.8)
    score1 += 0.2 * int(two_hop.get(0,0) > 0.8)
    cond1   = (score1 >= 0.5)

    # Exemplar #2 signature
    score2  = 0.5 * int(any(kw in features.lower()
                           for kw in ['logic program',
                                      'inductive logic programming',
                                      'ilp']))
    score2 += 0.3 * int(one_hop.get(0,0) > 0.7)
    score2 += 0.2 * int(two_hop.get(0,0) > 0.5)
    cond2   = (score2 >= 0.5)

    return cond1 or cond2
```

**Class 0 (Rule Learning)**

A publication belongs to the *Rule Learning* class if its text mentions "inductive logic programming," "meta-knowledge," or "hypothesis space," *or* if a large majority of its 1-hop and 2-hop citation neighbors also belong to Rule Learning.

Figure 13: Global explanation for TAGCora class 0 (Rule Learning): Top - the Python implementation as the OR of two exemplar signatures; Bottom - the human-readable description.

> **Global Explanations for TAGCORA**
>
> **Class 0 (Rule Learning)**
> Publications in "Rule Learning" contain keywords such as "inductive logic programming," "meta-knowledge," or "hypothesis space," and/or their immediate and 2-hop citation neighborhoods are predominantly Rule Learning papers.
>
> **Class 1 (Neural Networks)**
> Papers in "Neural Networks" mention terms like "neural network," "backpropagation," "neurons," or "activation functions," and/or their 1- and 2-hop neighbors are largely in the Neural Networks class; occasional focus on ICA or HMM further reinforces this label, while mentions of other paradigms reduce its likelihood.
>
> **Class 2 (Case-Based)**
> Case Based publications include phrases such as "case-based reasoning," "adaptation," or "planning," and frequently cite other Case Based works within one or two hops; secondary indicators include words like "learning" and "explanation."
>
> **Class 3 (Genetic Algorithms)**
> Genetic Algorithms papers feature keywords like "genetic algorithm," "evolutionary algorithm," or "genetic programming," and/or their 1-hop and 2-hop citation graphs are enriched with other Genetic Algorithms publications.
>
> **Class 4 (Theory)**
> "Theory" articles discuss Bayesian methods, classifiers, or decision trees, and/or they are frequently cited by, and cite, other Theory papers within one or two hops; a strong Theory neighborhood further disambiguates this class.
>
> **Class 5 (Reinforcement Learning)**
> Reinforcement Learning papers mention "reinforcement learning," "TD learning," or related terms such as "real-time," "online," or "neuro-evolution," and/or their direct and extended citations predominantly belong to the Reinforcement Learning class.
>
> **Class 6 (Probabilistic Methods)**
> Probabilistic Methods works include keywords like "Bayesian network," "Markov chain," "density estimation," "Gibbs sampling," or "MCMC," and/or at least 20–80% of their 1-hop and 2-hop neighbors are also Probabilistic Methods publications.

Figure 14: High-level, human-readable rule summaries for each class in the TAGCora dataset generated by our global explanation pipeline (see Fig. 13 for the detailed Python implementation and interpretation of the Rule Learning class).

> **Global Explanations for BAShapes**
>
> **Class 0 (Not in House)**
> Nodes not belonging to a house are characterized by having many (at least 6, often more than 8) neighbors that also don't belong to a house, and only a few (at most 2) neighbors in the house structure—reflecting their isolation from any house motifs.
>
> **Class 1 (Middle of House)**
> A "middle" node sits on the roof slope: it has exactly one tip neighbor, one base neighbor, and the other middle node as motif-neighbors, and at most two additional connections to non-house nodes.
>
> **Class 2 (Base of House)**
> A "base" node is directly connected to exactly one middle node and exactly one other base node, with no links to any other classes.
>
> **Class 3 (Tip of House)**
> A "tip" node occupies the apex of the house motif: it connects exclusively to two class-1 (middle) nodes and has no edges to any other class, forming the roof's peak.

Figure 15: High-level, human-readable rule summaries for each class in the BAShapes dataset generated by our global explanation pipeline.

> **Global Explanations for QUESTIONS**
>
> **Class 0 (Active Users)**
> Active users on Yandex Q are those whose immediate neighbors and 2-hop neighbors are also predominantly active, indicating engagement within a highly interactive community.
>
> **Class 1 (Inactive Users)**
> Inactive users exhibit feature profiles closely matching the exemplar node 11757 and have at least 50% of their immediate neighbors inactive, reflecting their embedding within a low-activity subnetwork.

Figure 16: High-level, human-readable rule summaries for each class in the Questions dataset generated by our global explanation pipeline.

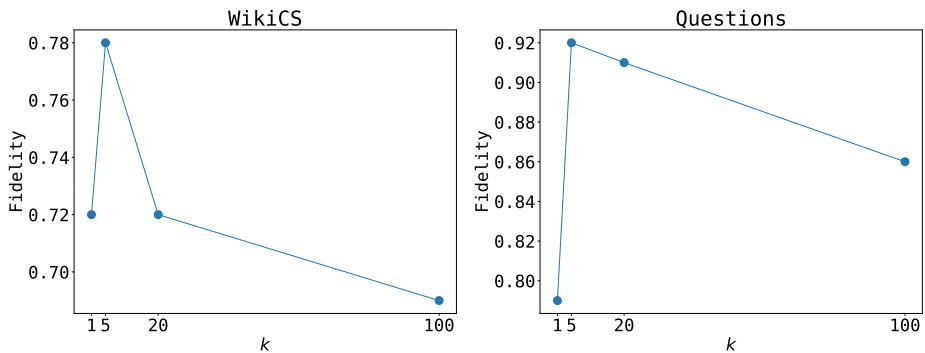

Figure 17: Effect of the Rev-$k$-NN $k$ on explanation fidelity for WikiCS and Questions. Each subplot plots mean fidelity over five runs as $k$ varies.

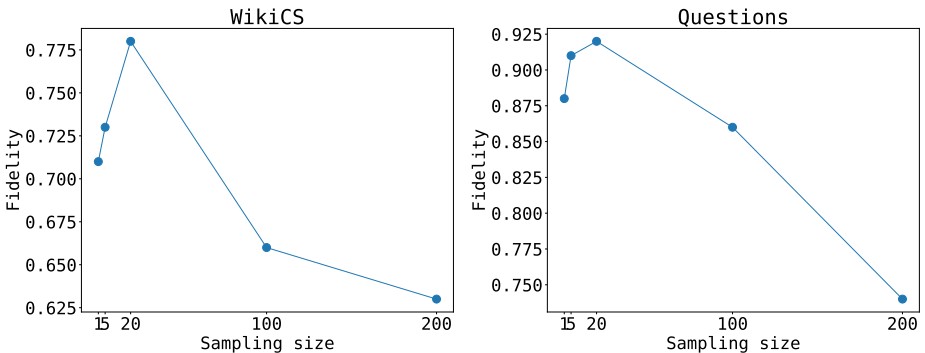

Figure 18: Impact of the training set sampling size on the explanation fidelity for WikiCS and Questions.

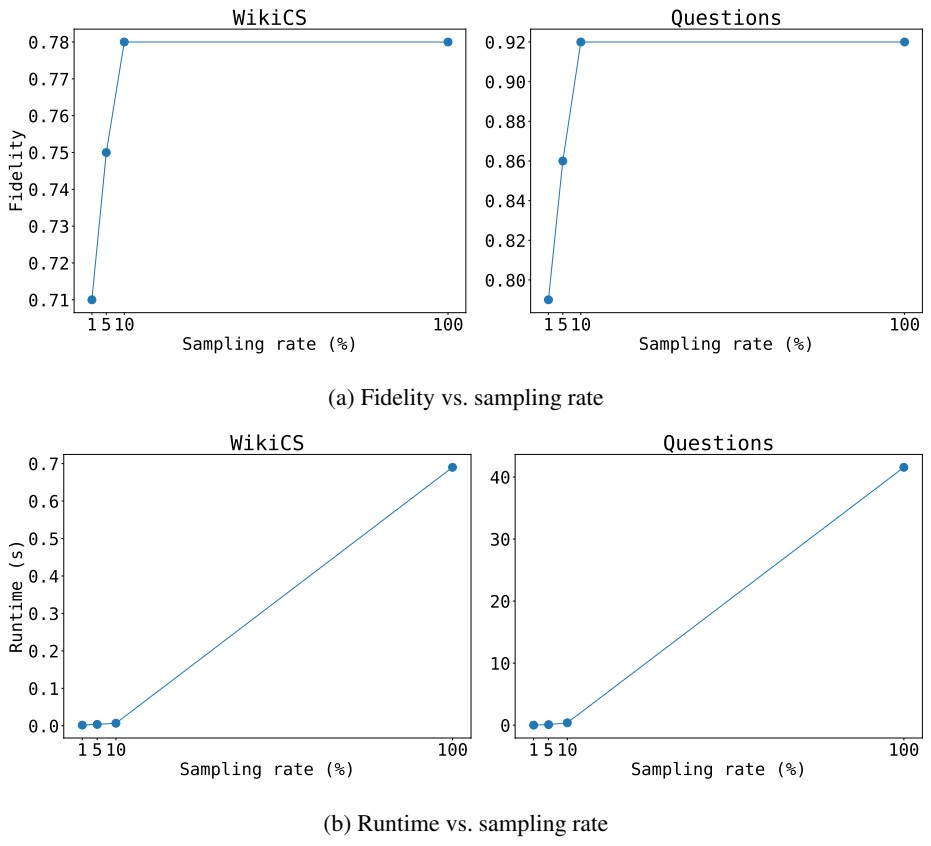

(a) Fidelity vs. sampling rate

(b) Runtime vs. sampling rate

Figure 19: Trade-off between runtime and fidelity as a function of sampling rate for WikiCS and Questions: (a) Fidelity as a function of sampling rate; (b) Exemplar selection runtime versus sampling rate.

