# OpenReview forum: "GnnXemplar: Exemplars to Explanations - Natural Language Rules for Global GNN Interpretability"
_NeurIPS.cc/2025/Conference — NeurIPS 2025 oral_

### Official Review · Reviewer_PJzR · 2025-06-07

**Clarity:** 3
**Significance:** 3
**Originality:** 3
**Rating:** 5
**Confidence:** 3

**Summary:**

This paper presented GNNXEMPLAR, a novel framework for global-concept-based GNN explanation of node classification in large graph. To achieve this, the author first accesses the GNN’s embedding space to identify exemplar nodes. Finally, to generate human-understandable explanations, they leverage LLMs to distill the defining characteristics of each exemplar and its neighborhood into concise natural language rules. As a result, this approach enables global yet instance-relevant explanations grounded in learned graph structure and semantics.

**Questions:**

Please refer to the above weakness section for suggestions.

**Ethical Concerns:**

["NO or VERY MINOR ethics concerns only"]

**Final Justification:**

I recommend a score of **5**. The concerns of other reviewers mainly lie in providing additional analyses, while the core claims of the paper are already supported by sufficient arguments. My main confusion is also solved: the authors have shown that GNNXEMPLAR can indeed run on small graphs like MUTAG (albeit with 68% fidelity versus GraphTrail’s 87%) and have clearly defined its intended domain—large, high-dimensional graphs without obvious motif structure. Overall, the method’s novelty and clear presentation are strong points, though there is an empirical limitation on small graphs.

**Limitations:**

yes

**Quality:**

3

**Strengths And Weaknesses:**

**Strengths:**
- The paper provides a fresh perspective on global-concept-based GNN explanation, with sufficient theoretical analysis. It is interesting to use LLM to extract rules.
- The method shows significant improvements in fidelity, scalability, and human interpretability across diverse benchmarks.
- The paper is well-written.

**Weaknesses:**
- GraphTrail can be compared on graph datasets with discrete node labels, and the same applies to MAGE. GNNs can transform discrete node labels into continuous embeddings. So, why can't GNNXEMPLAR be applied?

---

> ### Author Rebuttal · Authors · 2025-07-31
>
> We thank the reviewer for the constructive feedback. Please find below our response to the query made during the review. We remain open for any further discussion.
>
> ---
> 1. > **GraphTrail can be compared on graph datasets with discrete node labels, and the same applies to MAGE. GNNs can transform discrete node labels into continuous embeddings. So, why can't GNNXEMPLAR be applied?**
>
>  Indeed, GNNXEMPLAR can be applied to datasets like MUTAG used in GraphTrail. However, small graphs with discrete node labels are not the primary focus of our work, as several strong explainers (e.g., GraphTrail, MAGE) already perform well in that setting. Our goal is to address a more challenging and underexplored problem: explaining predictions in complex graphs with rich, high-dimensional node and edge attributes. For a detailed discussion of the associated challenges, please refer to lines 50–74 in our manuscript.
>
> Nonetheless, we conducted the requested experiment. We ran our method using the pre-trained GCN from GraphTrail (84% accuracy) and obtained a fidelity of 68%, compared to GraphTrail’s 87%. While GNNXEMPLAR correctly surfaced relevant substructures (e.g., NO2 groups), it fell short of capturing the precise motif-level patterns needed for high-fidelity explanation in this setting. For instance, here is an example output for the mutagenicity class:
>
> ```
> Class: Mutagenic
> The graph is classified mutagenic if it has at least 12 nodes, at least 12 carbon nodes, at least 1 nitrogen node, at least 2 oxygen nodes, at least 1 aromatic ring, and fewer than 2 halogen atoms (Cl, Br, F).
> These features are common among the class, while the presence of multiple halogen atoms is more characteristic of non-mutagenic graphs."
> ```
>
> These explanations are chemically meaningful but less granular than those recovered by GraphTrail, which is tailored to detect frequent subgraph motifs.
>
> This result highlights a broader point: `in small graphs where predictions hinge on discrete, recurring structural patterns, motif-centric methods like GraphTrail are more appropriate. In complex real-world networks characterized by high-dimensional, continuous features and lacking clear motif structures, such methods tend to fail.` GNNXEMPLAR is specifically designed for this latter setting, which remains underexplored. Thus, we believe this underperformance does not dilute the contribution of GNNXEMPLAR, but rather clarifies its intended use case.

---

> > ### Comment · Reviewer_PJzR · 2025-08-01
> >
> > Thank you for the detailed response and the additional experiment. My concerns have been addressed, and I will be maintaining my "accept" recommendation. The discussion regarding the method's performance on this aspect is an important one. I believe the paper would be significantly strengthened by including this analysis, as it clarifies the specific contributions and intended applications of your work for future researchers.

---

> > > ### Author Response · Authors · 2025-08-03
> > >
> > > We will certainly add the discussion presented in our rebuttal to the revised version. We thank Reviewer PJzR for the constructive suggestions and support for our work.
> > >
> > > **We also noted that your Accept rating no longer appears on the review. Could you kindly look into this?** The average rating of our work has also dropped to 4.33 from 4.5 since your rating stopped being visible on openreview.

---

> > > > ### Comment · Reviewer_PJzR · 2025-08-04
> > > >
> > > > My rating has always been 5 and has never been changed.

---

> > > > > ### Comment · Area_Chair_XM1x · 2025-08-05
> > > > >
> > > > > I can see that the rating is still 5: Accept.
> > > > >
> > > > > Your AC.

---

### Official Review · Reviewer_pgJY · 2025-06-10

**Clarity:** 4
**Significance:** 3
**Originality:** 4
**Rating:** 5
**Confidence:** 4

**Summary:**

The manuscript proposes a new framework for GNN explainability built on several main ideas: i) use of exemplars for global explanations (representative nodes), ii) application of an LLM to generate a disjunction of exemplar-specific rules for global explanations, and iii) using ideas from k-nearest neighbor and clustering methods to select exemplars.

**Questions:**

- Q1: [W1] The definitions are somewhat vague. Specifically, Definitions 3 and 4 are high level descriptions of an idea rather than concrete definitions. Can you provide more concrete definitions for the exemplar and signature?
- Q2: [W2] It would be helpful to comment on how the results of Theorem 2 would change if an estimate of Rev-k-NN is used (as in practical implementations here) instead of the actual Rev-k-NN.
- Q3: I wonder if a comparison with instance-level baselines such as GNNExplainer and PGExplainer on synthetic datasets such as BA Shapes can be provided as follows. Since the aforementioned explainers are instance-level, they need to be transformed into global explainers. Can they be combined with the LLM idea in this work, such that the explanation outputs of GNNExplainer and PGExplainer are input to the LLM to output the global rule? If so, how would the accuracy compare with the proposed explainer?

**Ethical Concerns:**

["NO or VERY MINOR ethics concerns only"]

**Final Justification:**

As mentioned in my response to the author's rebuttal, the responses have addressed all of my questions. I believe this is a novel and interesting work, and the manuscript is well-written.

With respect to the review score, I have maintained my evaluation at an Accept, which reflects the manuscript’s high quality and merit. However, I also believe it closely approaches the threshold for a Strong Accept.

**Quality:**

4

**Strengths And Weaknesses:**

Strengths:

- S1: The idea of using exemplars for global explanations is novel and quite interesting.
- S2: The manuscript complements the novel idea of using exemplars with several key innovations which enables a practical approach to exemplar generation including use of reverse k-means for exemplar selection, use of LLMs to generate the logical proposition for exemplar-specific rules, and the way the LLM is used to generate the Python code rather than a natural language explanation directly. Each of these ideas are novel and crucial in constructing an implementable and practical framework.
-  S3: The manuscript is well-written and contains useful examples and intuitive explanations, e.g., the example after Definition 4 is helpful.
- S4: The theoretical results, while straightforward to prove, novel and insightful. For instance, Theorem 2, providing a lower bound on the performance of the greedy algorithm is a useful result.
Weaknesses:
- W1: The definitions are somewhat vague. Specifically, Definitions 3 and 4 are high level descriptions of an idea rather than concrete definitions.
- W2: It would be helpful to comment on how the results of Theorem 2 would change if an estimate of Rev-k-NN is used (as in practical implementations here) instead of the actual Rev-k-NN.

---

> ### Author Rebuttal · Authors · 2025-07-31
>
> We appreciate the reviewer's constructive feedback and have revised the manuscript accordingly. We would these changes address the concerns raised.
>
> ----
>
> > **q1 / w1: The definitions are somewhat vague. Specifically, Definitions 3 and 4 are high level descriptions of an idea rather than concrete definitions. Can you provide more concrete definitions for the exemplar and signature?**
>
> To improve precision and clarity, we will supplement our definitions with the following mathematical formulations:
>
> **Definition 3 (Exemplars)**
> A node $e \in \mathcal{V}_{\text{tr}}$ is called an exemplar for the class $c \in \mathcal{Y}$ if $|R_e| \geq \tau$, where
>
> $$
> R_e = \left\\{
> v \in \mathcal{V}_{\text{tr}} \\ \middle|
> \begin{aligned}
>     &\Phi(e) = c, \\\\
>     &\Phi(v) = c, \\\\
>     &d(z_e, z_v) \leq \delta
> \end{aligned}
> \right\\}
> $$
> where $z_v \in \mathbb{R}^d$ is the GNN-learned embedding of node $v$, $d(z_e, z_v)$ is some distance function, $\delta \in \mathbb{R}$ is a distance threshold, and $\tau \in \mathbb{N}$ is a minimum population size hyperparameter. As we will see later,  the thresholds will automatically be discovered from the data by exploiting the $k$-nearest neighbor relationship along with a greedy selection algorithm.
>
> **Definition 4 (Exemplar Signature)**
> The signature $\sigma_e$ of exemplar $e$ from class $c$ is a boolean function $f(\pi_1(v), \pi_2(v), \dots, \pi_k(v)) \in \{0,1\}$, such that, with a high probability, $\forall v\in R_e, \sigma_e(v) = 1$, iff $\Phi(v)=c$. Here, $\pi_i$ are the predicates that evaluate to Boolean conditions over interpretable properties such as:
> - Node features (e.g., $x_v[\texttt{age}] > 30$)
> - Local structure (e.g., $\texttt{degree}(v) \geq 3$)
> - Neighborhood composition (e.g., $\texttt{fractionClass}(v, \  \texttt{hop}=1, \\ c') > 0.6$)
>
> >**q2/w2. It would be helpful to comment on how the results of Theorem 2 would change if an estimate of Rev-k-NN is used (as in practical implementations here) instead of the actual Rev-k-NN.**
>
> Let $\widetilde{\text{Rev}}{\text{-}k\text{-NN}}(v)$ denote the approximate reverse k-NN set of a node $v$, and $\widetilde{\Pi}(A)$ denote the approximate coverage function computed using these sets. The greedy algorithm in Theorem 2 provides a $(1 - 1/e)$-approximation **to the maximization of $\widetilde{\Pi}(A)$**, since the proof relies solely on submodularity and monotonicity of the input function—both of which are preserved under the approximation (as the union of approximate sets still yields a valid set function).
>
> To relate this back to the *true* optimum $\Pi(A^*)$, we leverage the uniform error bound from Lemma 1 (Appendix A.3), which ensures that for any set $A$:
>
> $$
> |\widetilde{\Pi}(A) - \Pi(A)| \leq \theta \quad \text{with high probability (≥ $1 - \delta$)}.
> $$
>
> Let $\widetilde{A}$ be the set returned by greedy maximization over $\widetilde{\Pi}$, and $A^*$ be the optimal set under the true objective $\Pi$. The greedy guarantee gives:
>
> $$
> \widetilde{\Pi}(\widetilde{A}) \geq \left(1 - \frac{1}{e} \right)\widetilde{\Pi}(A^*).
> $$
>
> By the uniform bound, we know $\widetilde{\Pi}(A\^{\*}) \geq \Pi(A\^{\*}) - \theta$, so:
>
> $$
> \widetilde{\Pi}(\widetilde{A}) \geq \left(1 - \frac{1}{e} \right)(\Pi(A\^{\*}) - \theta)
> = \left(1 - \frac{1}{e} \right)\Pi(A\^{\*}) - \left(1 - \frac{1}{e} \right)\theta.
> $$
>
> **Thus, even when using approximate Rev-k-NN sets, the greedy algorithm achieves the following guarantee on estimated coverage:**
>
> $$
> \boxed{
> \widetilde{\Pi}(\widetilde{A}) \geq \left(1 - \frac{1}{e} \right)\Pi(A^*) - \left(1 - \frac{1}{e} \right)\theta
> }
> $$
>
> We will include this clarification in the revised version. We also have a supporting experiment in the paper for the same. Please refer Appendix A.6 "Sampling rate for exemplar selection".
>
> > **q3. I wonder if a comparison with instance-level baselines such as GNNExplainer and PGExplainer on synthetic datasets such as BA Shapes can be provided as follows. Since the aforementioned explainers are instance-level, they need to be transformed into global explainers. Can they be combined with the LLM idea in this work, such that the explanation outputs of GNNExplainer and PGExplainer are input to the LLM to output the global rule? If so, how would the accuracy compare with the proposed explainer?**
>
> In principle, instance-level explainers like GNNExplainer and PGExplainer could be used to generate local explanations for a collection of representative nodes, and their outputs could then be aggregated and passed to an LLM to produce a global explanation. However, there are fundamental challenges with this approach:
>
> 1. **Local explanations do not guarantee semantic convergence:**
>    Local explainers aim to explain the prediction on a specific instance (node/graph) by identifying the most relevant subgraph or features. There is no incentive or regularization encouraging consistency across local explanations for nodes in the same class. As a result, local explanations often differ substantially across instances—even within a single class—due to variations in neighborhood structure or optimization noise.
>
>    Consequently, when such varied and potentially inconsistent explanations are provided to an LLM, the output global rule may be incoherent, fragmented, or overfit to specific patterns that do not generalize.
>
> 2. **Our method ensures explanation consistency by design:**
>    In contrast, GNNXEMPLAR selects exemplars based on the GNN’s learned semantic space and associates each with a *population*—a reverse-kNN set of nodes whose representations are similar. This promotes semantic alignment across the class and ensures that the LLM receives a *coherent and representative subset* from which it can learn generalizable rules. Moreover, this design enables the LLM to learn global discriminative patterns rather than instance-specific artifacts.
>
> **Experiment:** To empirically support this argument, we conducted the experiment as suggested by the reviewer. We applied PGExplainer on the BA-Shapes dataset to generate local explanations for nodes from each class. These explanations were then passed to an LLM with the goal of producing global rules.
>
> Despite careful filtering of spurious explanations (e.g., restricting to explanations with <30 edges), the resulting fidelity was only 46%.

---

> > ### Comment · Reviewer_pgJY · 2025-07-31
> >
> > I thank the authors for their comprehensive response. The response addresses all of my questions. As mentioned in my original review, I believe this is a novel and interesting work, and the manuscript is well-written.
> >
> > With respect to the review score, I maintain my evaluation at an Accept, which reflects the manuscript’s high quality and merit. However, I also believe it closely approaches the threshold for a Strong Accept.

---

### Official Review · Reviewer_b9jp · 2025-06-24

**Clarity:** 4
**Significance:** 3
**Originality:** 3
**Rating:** 4
**Confidence:** 3

**Summary:**

This paper proposes GNNXEMPLAR, a novel global explanation framework for Graph Neural Networks (GNNs) applied to node classification tasks. The method draws inspiration from exemplar theory in cognitive science, selecting representative nodes (exemplars) in the GNN embedding space and generating global explanations via natural language rules derived from their neighborhoods. Exemplar selection is formulated as a coverage maximization problem using reverse k-nearest neighbors, solved via a greedy algorithm with submodular approximation guarantees. The natural language rule generation employs a self-refining prompting strategy leveraging large language models (LLMs). Extensive experiments across multiple homophilous and heterophilous benchmarks demonstrate GNNXEMPLAR's superiority in fidelity, scalability, and human interpretability.

**Questions:**

See above weaknesses.

**Ethical Concerns:**

["NO or VERY MINOR ethics concerns only"]

**Limitations:**

YES

**Quality:**

3

**Strengths And Weaknesses:**

Strengths:
1. The paper addresses a clear gap in GNN interpretability research by focusing on global explanations for node classification on large, real-world graphs with high-dimensional node attributes. The shift from motif-based to exemplar-based explanations is both novel and well-motivated.
2. The combination of reverse k-NN for exemplar selection and LLM-driven self-refinement for natural language rule discovery is highly innovative. The theoretical formulation, including NP-hardness proofs and submodularity-based greedy optimization, demonstrates methodological rigor.
3. The experiments are extensive, covering a diverse set of datasets, GNN architectures, and both homophilous and heterophilous graphs. Ablation studies clearly demonstrate the contribution of each module.

Weaknesses:
1. The explanation of GNNXEMPLAR is actually "mimicking the output behavior of GNN", but it does not really restore the decision logic of the internal mechanism of GNN. And the whole method is highly dependent on whether GNN has learned embeddings with semantic distinction ability during training. Once the GNN training effect is poor or overfitting or underfitting, the quality of Exemplar selection and explanation rules will be affected.
2. The comparison focuses primarily on global explainers adapted from graph classification; comparisons to more recent feature-attribution-based or counterfactual node explainers could provide additional perspective（GOAt，COMRECGC）.
- [1] Lu S, Mills K G, He J, et al. GOAt: Explaining graph neural networks via graph output attribution[J]. arXiv preprint arXiv:2401.14578, 2024.
- [2] Fournier G, Medya S. COMRECGC: Global Graph Counterfactual Explainer through Common Recourse[J]. arXiv preprint arXiv:2505.07081, 2025.
3. The current framework operates only on the embedding space and lacks full visibility into the internal feature-topology interactions of GNNs, limiting its mechanistic interpretability.

---

> ### Author Rebuttal · Authors · 2025-07-30
>
> We really appreciate your detailed comments. They have improved our work.
>
> ----
> > **w1a. The explanation of GNNXEMPLAR is actually "mimicking the output behavior of GNN", but it does not really restore the decision logic of the internal mechanism of GNN.**
>
> **Ans:** We agree that GNNXEMPLAR does not recover the internal decision logic of the GNN in a mechanistic sense. We will clearly note this limitation in the revised version.
>
> That said, GNNXEMPLAR is not simply mimicking GNN outputs through a detached surrogate model. Its exemplar selection operates directly over the GNN’s latent embedding space, leveraging reverse k-nearest neighbor (rKNN) relationships to identify points that capture the semantic structure induced by the model. The resulting explanations reflect how the GNN partitions and organizes the embedding space to encode task-relevant decision making.
>
> This connection between embedding distances with GNN prediction logic is now supported by the ICML 2025 paper `WILTing Trees: Interpreting the Distance Between MPNN Embeddings`, which shows that:
> > *“MPNN distances after training are aligned with the task-relevant functional distance of the graphs and that this is key to the high predictive performance of MPNNs.”*
>
> They further observe that MPNNs align embeddings more with *functional* than *structural* similarity.
>
> >**w1b ..And the whole method is highly dependent on whether GNN has learned embeddings with semantic distinction ability during training. Once the GNN training effect is poor or overfitting or underfitting, the quality of Exemplar selection and explanation rules will be affected.**
>
> The reviewer is again correct in observing that GnnXemplar depends on the GNN learning embeddings with semantic separation.  The exemplar selection assumes that the GNN organizes the latent space meaningfully. Therefore, if the GNN underfits the data, and the embeddings are effectively noise, GnnXemplar will struggle to extract meaningful exemplars and subsequent explanations.
>
> However, the situation is different in the case of overfitting. When the GNN picks up spurious correlations and learns incorrect patterns, GnnXemplar is able to surface these faithfully, providing valuable diagnostic insight into the model’s flawed decision process. This was demonstrated in our case study in Section 4.6, where class imbalance led the GNN to learn an incorrect oversimplified rule. GnnXemplar followed the GNN's decision logic which revealed that the GNN's logic was misaligned with the intended class semantics. Such insights underscore GnnXemplar's strength as a failure analysis tool as it faithfully surfaces what the model has actually learned, regardless of whether that learning is correct wrt. the ground truth.
>
> >**w2.  The comparison focuses primarily on global explainers adapted from graph classification; comparisons to more recent feature-attribution-based or counterfactual node explainers could provide additional perspective（GOAt，COMRECGC).**
> > - [1] **Lu S, Mills K G, He J, et al. GOAt: Explaining graph neural networks via graph output attribution[J]. arXiv preprint arXiv:2401.14578, 2024.**
> > - **[2] Fournier G, Medya S. COMRECGC: Global Graph Counterfactual Explainer through Common Recourse[J]. arXiv preprint arXiv:2505.07081, 2025.**
>
> We thank the reviewer for highlighting these recent methods. We agree that they are valuable contributions to the explainability landscape but note that both **GOAt** and **COMRECGC** target substantially different tasks and assumptions, making a direct comparison to **GNNXEMPLAR** unsuitable:
>
> 1. **GOAt** is a *local* explainer that attributes predictions to individual nodes/edges/features via analytical decomposition. In contrast, **GNNXEMPLAR** is a *global* explainer that derives class-level rules using exemplar populations and natural language. The methods differ in scope (local vs. global), output (numeric masks vs. symbolic rules), and mechanism (per-instance attribution vs. population-based abstraction). Aggregating GOAt's outputs into coherent global rules is hence non-trivial. Thus, a direct comparison is not meaningful due to the fundamental differences in objective, output, and interpretability granularity. Nonetheless, we will definitely cite this work and discuss the differences.
>
> 2. **COMRECGC** is a *counterfactual global explainer* operating under assumptions that do not align with our problem setting:
>    - It relies on *graph edit distance*, which is undefined for graphs with continuous, high-dimensional node features—common across our benchmarks.
>    - It assumes recurring structural motifs across input graphs, while **GNNXEMPLAR** is explicitly designed for settings where such motifs are rare or absent (e.g., citation or social networks).
>    - COMRECGC focuses on *recourse-based explanations*, requiring binary "accept" and "reject" classes. This framing does not generalize well to multiclass node classification, where defining a privileged positive class introduces arbitrary asymmetry and limits interpretability.
>    - Despite the above differences, we attempted to run COMRECGC on our datasets, but encountered implementation challenges due to the reasons above—particularly the absence of recurring substructures and the inapplicability of GED in our setting.
>    - Finally, we note that **COMRECGC was published in ICML just weeks back**, which is **two months after the NeurIPS submission deadline**. Nonetheless, we will ensure it is properly cited and discussed in the revised version.
>
> >**w3. The current framework operates only on the embedding space and lacks full visibility into the internal feature-topology interactions of GNNs, limiting its mechanistic interpretability.**
>
> This connects to our response to w1a. Once again, we accept this limitation and shall explicitly mention it in the final version of our paper.

---

> > ### Comment · Reviewer_b9jp · 2025-08-05
> >
> > Thanks for the rebuttal. I would like to keep my positive rating on the paper.

---

### Official Review · Reviewer_R8Cq · 2025-07-02

**Clarity:** 3
**Significance:** 3
**Originality:** 3
**Rating:** 5
**Confidence:** 4

**Summary:**

The paper introduces an approach for model-level explanations of GNN
predictions focused on node classification in large graphs, where
motif-based explanations are less likely to be effective (and are
expensive to compute). The approach combines the identification of
exemplars node characterizing the model behavour for each class with
the extraction of textual explanations via LLM prompting.

**Questions:**

- How does the reverse k-NN selection problem differs from coreset identification?

- Can you clarify what precision/recall/f1 represent? are these referring to the classification performance on the explainer? what about the one of the original GNN?

- BAShapes: how are labels incorporated? as additional attributes?

**Ethical Concerns:**

["NO or VERY MINOR ethics concerns only"]

**Final Justification:**

The authors addressed my concerns in a satisfactory way in the rebuttal. I expect the main aspects of the feedback to be incorporated in a revised version of the manuscript.

**Limitations:**

The work does include a paragraph on limitations. However I see a main limitation that is not discussed, with is the strong bias in guiding the LLM to extract explanations, and the difficulty in
adapting it for truly topological patterns.

**Quality:**

3

**Strengths And Weaknesses:**

# Strengths

- The approach addresses a poorly explored task, which is model-level
explanation in node classification.

- The proposed solution is sensible, well-motivated and explained.


# Weaknesses

- The problem of identifying exemplars sound very similar to the one of
coreset selection. It would be useful to understand why the authors
could not use this well-developed framework and how the proposed
solution differs from it.

- The approach sounds like a prototype-based explainability method. I
guess the PAGE approach [1] is a very relevant piece of work that
should be cited and compared with (even if the focus on graph-level
explainability remains).

- The key element allowing LLMs to generate interpretable explanations
is the summary of the node neighborhood. A very strong bias is
introduced there, as the topology is flattened into sets of l-hop
neighbors, and the attributes are treated independently. This is
reflected in the explanations being generated. I have the impression
that this approach would have problems when topology and attribute
relationships play a stronger role in the prediction.

# Minor points

- For the sake of consistency, I would use lowercase non-calligraphic letters for individual classes (calligraphic letters indicate sets in the paper)

- Phi is used in Def. 4 before being introduced.

[1] Shin, Y.-M., Kim, S.-W., Yoon, E.-B., & Shin, W.-Y. (2022). Prototype-Based Explanations for Graph Neural Networks (Student Abstract). Proceedings of the AAAI Conference on Artificial Intelligence, 36(11), 13047-13048. https://doi.org/10.1609/aaai.v36i11.21660

---

> ### Author Rebuttal · Authors · 2025-07-30
>
> We thank the reviewer for the constructive feedback on our work. We hope the proposed revisions and clarifications would address the concerns raised.
>
> -----
>  > **q1 and w1. The problem of identifying exemplars sound very similar to the one of coreset selection. It would be useful to understand why the authors could not use this well-developed framework and how the proposed solution differs from it.**
>
> **Ans.** Thank you for the thoughtful observation. While our exemplar selection shares surface similarities with coreset selection—both involve selecting representative data points—their goals, design principles, and outputs differ fundamentally:
>
> 1. **Objective Misalignment:**
>    Coreset selection is primarily geared toward preserving predictive performance. It identifies a subset of training data such that learning on this subset yields comparable accuracy to learning on the full dataset. This means it often prioritizes samples near the decision boundary—those most informative for minimizing training loss.
> In contrast, our goal is global interpretability, not prediction. We seek exemplars that capture the semantic diversity of a class in the GNN embedding space. The goal is to find prototypical points—not ambiguous ones near the decision boundary—that characterize dense, consistent regions of class semantics.
>
> 2. **Coverage vs. Loss Optimization:**
>    Coreset methods evaluate the *joint effect* of selected points on the training objective. Our method optimizes a *submodular coverage function* (Eq. 2) over reverse k-nearest neighbors, ensuring that diverse regions of each class are well-represented. This leads to greater interpretability, even when using very few exemplars (often 2–3 per class), which would be insufficient for a coreset due to underfitting risk.
>
> 3. **Exemplar Populations and Interpretability:**
>    Our method returns not only exemplars but also their *populations*—i.e., the set of nodes each exemplar represents via reverse k-NN. This is critical for generating *explanation rules* (signatures), as the rule is learned over the exemplar’s population. Coresets do not provide such mappings, and thus cannot support this explanation mechanism.
>
> 4. **Interpretability Trade-off:**
>    Since coreset points may lie near decision boundaries, they tend to be ambiguous or noisy—counterproductive for human interpretation. Our approach deliberately avoids such regions to ensure that selected exemplars yield *clear, faithful, and symbolic* global explanations.
>
>
> > **w2. The approach sounds like a prototype-based explainability method. I guess the PAGE approach [1] is a very relevant piece of work that should be cited and compared with (even if the focus on graph-level explainability remains).**
>
> **Ans:** Thank you for pointing us to PAGE. While it is a relevant prototype-based explainer, it is fundamentally designed for *graph-level* tasks on small datasets with *discrete* features, and does not scale to the *node classification* setting on *large graphs with high-dimensional or continuous node features*.
>
> We attempted to run PAGE on our benchmarks using the official codebase but encountered crashes on all datasets. The core reasons are:
>
> 1. **Scalability Issues:**
>    PAGE performs prototype search using a Cartesian product across graphs, resulting in a combinatorial explosion (O(n³) for k=3), which is computationally infeasible on large graphs.
>
> 2. **Feature Constraints:**
>    The method assumes one-hot or categorical node features. Most of our datasets use continuous or bag-of-words features, making PAGE incompatible without significant reengineering.
>
> 3. **Subgraph Matching Limitations:**
>    Even if subgraphs are extracted, evaluating fidelity becomes intractable due to the difficulty of subgraph isomorphism over continuous features.
>
> 4. **Rigid Output Modality:**
>    PAGE produces subgraph-based explanations, which—as we discuss in the paper—become unwieldy and less interpretable in large, real-world graphs.
>
> We will cite PAGE in the related work and include a brief discussion outlining these limitations in the revised version.
>
> > **w3 and Limitations. The key element allowing LLMs to generate interpretable explanations is the summary of the node neighborhood. A very strong bias is introduced there, as the topology is flattened into sets of l-hop neighbors, and the attributes are treated independently. This is reflected in the explanations being generated. I have the impression that this approach would have problems when topology and attribute relationships play a stronger role in the prediction.**
>
> **Ans:** We agree with the reviewer that flattening the neighborhood structure into a set of aggregated l-hop neighbors limits the LLM’s ability to capture fine-grained structure-attribute relationships. However, we clarify that GnnXemplar is not inherently limited to aggregated prompts. This design choice was made for two practical reasons:
>
> 1. **LLM context constraints:** Real-world graphs often have large or dense neighborhoods, making it infeasible to encode full adjacency structures within the LLM’s context window. Aggregated representations offer a scalable compromise for such settings.
> 2. **Nature of real-world graphs:** As motivated in the paper, most real-world node classification graphs (e.g., citation, product, or social networks) do not contain clear, repeating motifs where explicit structure is predictive. In such cases, preserving exact topological detail is unnecessary and even distracting.
>
> That said, the self-refine pipeline is flexible. In domains where explicit structure-function relationships are crucial (e.g., molecular graphs), users can customize the initial prompt format to include explicit neighborhood structure and features. For instance, if the graph is small and contains rich structural motifs, the user can override the default aggregation and feed detailed structural information directly. The LLM will then learn to generate structure-aware explanations accordingly.
>
> To further demonstrate that GnnXemplar can operate effectively even in settings where structure-attribute synergy is essential, we designed a synthetic dataset where flattened neighborhood representations are insufficient, and neither structure nor features alone suffice for explanation.
>
> **Dataset Design**:
> Node features consist of two colors: red and blue. Class labels are defined as follows:
>
> - **Class 0**: A node is in class 0 if it is red and part of a triangle where others are also red.
> - **Class 1**: A node is in class 1 if it is red and it belongs to a diamond structure (a 4-cycle), regardless of the colors of the other nodes in the house.
> - **Class 2**: A node is in class 2 if it is blue and it belongs to a diamond structure (a 4-cycle), regardless of the colors of the other nodes in the house.
> - **Class 3**: Any node not satisfying the above conditions.
>
> To make the task non-trivial, we deliberately injected confounding impurities, such as:
> - Blue nodes in triangles.
> - Red nodes connected to two red neighbors that don’t form a triangle.
> - Chains of length four to confound diamonds.
>
> Clearly, correct classification here requires joint reasoning over both features and topology. Flattened prompts or purely structural/feature-based reasoning would be insufficient.
>
> GnnXemplar, when run with prompts that preserved structural detail, achieved 93% fidelity.
>
> The generated explanations were:
> ```
> Class 0:
> A node belongs to class 0 if it is red, part of a triangle, not part of a diamond, has two or three connections to other nodes.
>
> Class 1:
> A node belongs to class 1 if it is red and part of a diamond shape, but not part of a triangle shape.
>
> Class 2:
> A node belongs to class 2 if it is blue and is part of a diamond shape.
>
> Class 3:
> Class 3 nodes are typically red and part of a triangle, but can also be blue and part of a triangle; nodes that are part of a diamond shape are less likely to belong to this class.
> ```
>
> These explanations demonstrate that GnnXemplar can effectively discover and express complex class-level patterns even when fine-grained structure-feature relationships are required.
>
> We hope this evidence reassures the reviewer that GnnXemplar is not limited to flattened neighborhood reasoning and can handle datasets with strong structure-function dependencies when prompts are crafted accordingly.
>
> > **q2. Can you clarify what precision/recall/f1 represent? are these referring to the classification performance on the explainer? what about the one of the original GNN?**
>
> **Ans:** All the metrics are calculated against the GNN's predictions, not the ground truth.
>
> > **q3. BAShapes: how are labels incorporated? as additional attributes?**
>
> **Ans:** In BAShapes, as in all our datasets, we do **not** provide ground-truth labels to the LLM. The LLM only receives the **GNN’s predicted labels**, as stated in lines 257–259. By using predicted labels rather than true ones, we ensure that the LLM explanations reflect the **GNN’s learned behavior**, not the dataset’s ground-truth semantics.
>
> > **Minor points**
>
> Thank you for pointing them out. We will address them.
>
> [1] Shin, Y.-M., Kim, S.-W., Yoon, E.-B., & Shin, W.-Y. (2022). Prototype-Based Explanations for Graph Neural Networks (Student Abstract). Proceedings of the AAAI Conference on Artificial Intelligence, 36(11), 13047-13048

---

> > ### Comment · Reviewer_R8Cq · 2025-08-04
> >
> > Thanks a lot for your detailed feedback, which did address the main points I had raised. I encourage the authors to incorporate the feedback (related work for coresets and SAGE, experimental part for the synthetic experiment) to strengthen the contribution.

---

> > > ### Author Response · Authors · 2025-08-04
> > >
> > > We appreciate your feedback and are happy to have addressed your concerns. We will incorporate the suggestions from this discussion and thank you for your time and help in improving our paper.

---

### Decision · Program_Chairs · 2025-09-17

**Decision:**

Accept (oral)

**Comment:**

This paper presents GnnXemplar, a global explainer inspired by Exemplar Theory from cognitive science. GnnXemplar identifies representative nodes in the GNN embedding space, called exemplars, using reverse-kNN. Then, Exemplar Signatures are discovered through iterative self-refinement of LLM outputs, leveraging their strong reasoning abilities and established strengths in linguistic articulation. The pipeline of GnnXemplar is clearly described in Figure 1, and explanations are derived from the exemplars' neighborhoods.

Although local explanations for individual cases have been proposed, developing a global explainer to characterize an entire class has remained difficult. The validity of the proposed explainer has been demonstrated experimentally using eight datasets and three GNN architectures: GnnXemplar consistently showed the highest fidelity scores, while previous approaches often suffered from out-of-memory issues or subgraph isomorphism. In addition, a user study involving 60 participants showed a clear preference for textual explanations.

All the reviewers were positive about the paper’s technical novelty and theoretical depth, and they were satisfied with the experimental validations. As shown in Table 2 and Figure 3, the stability and superiority of GnnXemplar are clear (e.g., fidelity scores of 0.92 for both Citeseer and Questions). Although the reviewers were already positive before the authors’ rebuttals, the rebuttal comments further deepened their understanding of this paper.

This work demonstrates clear novelty by introducing a fundamentally new paradigm for global GNN interpretability: shifting from motif-based subgraph discovery to exemplar-based natural language rules. It exhibits strong technical contributions to the machine learning community, with theoretical guarantees (NP-hardness, submodularity, and approximation bounds), carefully designed algorithms (reverse k-NN exemplar selection with greedy optimization), and comprehensive experiments including ablation studies, robustness across GNN architectures, and a comprehensive human user study. Its broad significance to the NeurIPS community lies in bridging GNN interpretability and large language models. The approach not only advances methodological foundations but also provides practical insights for failure diagnosis and trustworthy AI, which are timely and critical issues in the community.

For these reasons, the paper will give great influence in explainable graph learning, attract wide interest across both theoretical and industrial aspects, and stimulate further exploration of exemplar and language-based interpretability paradigms. Therefore, I recommend that the paper should be accepted as an oral presentation.